# IMPLICIT DYNAMICAL FLOW FUSION (IDFF) FOR GENERATIVE MODELING

## ABSTRACT

Conditional Flow Matching (CFM) models can generate high-quality samples from a non-informative prior, but they can be slow, often needing hundreds of network evaluations (NFE). To address this, we propose Implicit Dynamical Flow Fusion (IDFF); IDFF learns a new vector field with an additional momentum term that enables taking longer steps during sample generation while maintaining the fidelity of the generated distribution. Consequently, IDFFs reduce the NFEs by a factor of ten (relative to CFMs) without sacrificing sample quality, enabling rapid sampling and efficient handling of image and time-series data generation tasks. We evaluate IDFF on standard benchmarks such as CIFAR-10 and CelebA for image generation, where we achieve likelihood and quality performance comparable to CFMs and diffusion-based models with fewer NFEs. IDFF also shows superior performance on time-series datasets modeling, including molecular simulation and sea surface temperature (SST) datasets, highlighting its versatility and effectiveness across different domains.

## 1 INTRODUCTION

Diffusion models have emerged as powerful tools for modeling complex, high-dimensional data by iteratively transforming random noise into meaningful information (Cachay et al., 2023; Myers et al., 2022). These models have demonstrated remarkable success in various domains, including high-quality image generation (Song et al., 2020a) and text production (Liu et al., 2024; Kim et al., 2019). However, their training process remains computationally intensive, often requiring hundreds of function evaluations (NFEs) to achieve high-quality results (Ho et al., 2020). While techniques such as DPM-solver (Lu et al., 2022a;b) and Denoising Diffusion Implicit Models (DDIMs) (Song et al., 2020b) have made strides in reducing NFEs for diffusion-based generative models, significant computational challenges persist.

In contrast to diffusion models, which parameterize the dynamics of noise-to-data transformation, Conditional Flow Matching (CFMs) directly parameterize vector fields governing this transformation (Liu et al., 2022; Albergo and Vanden-Eijnden, 2022; Albergo et al., 2023). CFMs leverage the change of variable theory in statistics to define conditional probability paths, enabling faster convergence during training compared to diffusion models (Lipman et al., 2022). However, generating high-quality samples with CFMs still requires hundreds of NFEs, making the process computationally expensive (Dao et al., 2023) and limiting the extension of CFM models to time-series data modeling, as NFEs scale with sequence length.

To address these challenges, we introduce a novel method to enhance the efficiency of CFMs, allowing for longer sampling steps without compromising the fidelity of the target distribution. Our approach draws inspiration from Hamiltonian Monte Carlo (HMC) algorithms (Neal, 2012a; Neal et al., 2011), which utilize the conservation properties of Hamiltonian dynamics to couple target and momentum variables. The key contributions of our work are as follows:

1. We propose Implicit Dynamical Flow Fusion (IDFF), which integrates a momentum term into the vector field of conditional flow models. Similar to how HMC reduces the number of samples required to reach the target distribution (Neal, 2012a; Neal et al., 2011), **IDFF reduces the NFE by a factor of 10 compared to traditional CFMs** while maintaining identical marginal

CelebA-256    LSUN    ImageNet-64    CelebA-64   CIFAR-10

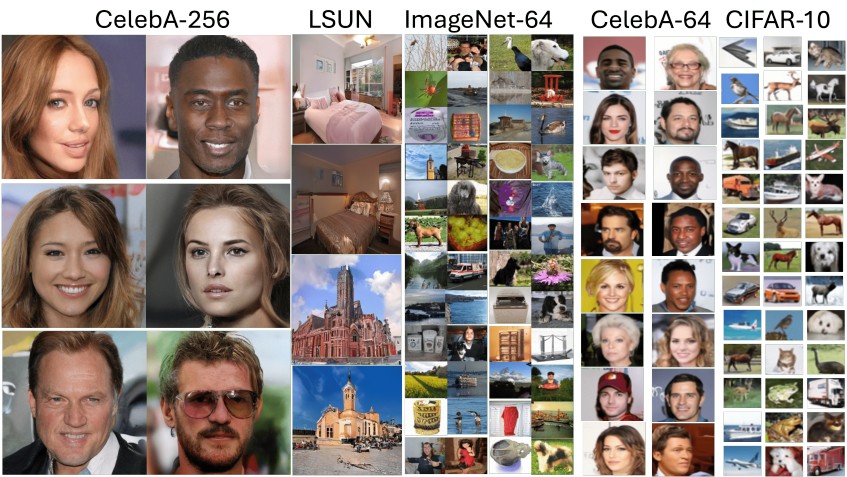

Figure 1: Image generation using IDFF (with NFE=10) applied to multiple datasets: CIFAR10, CelebA-64, ImageNet-64, LSUN-Bedroom, LSUN-Church, and CelebA-HQ. The quality of the generated images demonstrates IDFF's ability to capture and reproduce realistic visuals across varying levels of complexity and resolution. Additional samples and further analysis can be found in Appendix A.6.

distributions. Additionally, we introduce a training objective that directly aligns generated samples with actual data rather than solely focusing on matching conditional flows.

2. We validate IDFF through extensive experiments on image generation using datasets such as CIFAR-10 and CelebA. Additional results on ImageNet-64, CelebA-HQ, LSUN-Church, and LSUN-Bedroom are presented in Figure 1. Our models achieve a superior balance between computational cost and sample quality compared to both traditional diffusion models (e.g., DDPM (Ho et al., 2020), DDIM (Song et al., 2020b)) and current CFMs.

3. We demonstrate that reducing NFEs makes IDFF practical for other generation tasks, such as time-series modeling. We showcase IDFF's effectiveness in capturing complex dynamical systems, including 3D attractors, molecular dynamics (MD), and sea surface temperatures (SST). Our results indicate that IDFF surpasses other methods in sample quality for MD and SST datasets while maintaining computational efficiency, with NFE $\leq 5$ for each generated sample.

## 2   BACKGROUND

**Score-based generative models.** Let $p_{\text{data}} = \{\mathbf{x}^1, \mathbf{x}^2, \ldots, \mathbf{x}^N\}$, where $\mathbf{x}^i \overset{\text{iid}}{\sim} p(\mathbf{x})$ and $\mathbf{x}^i \in \mathbb{R}^d$ for all $i = 1, 2, \ldots, N$. We aim to build a generative model for this dataset given empirical samples. Moving forward, we suppress the superscript on each $\mathbf{x}^i$.

Score-based models (Song and Ermon, 2019) represent a broad class of diffusion models that describe a continuous-time stochastic process characterized by a stochastic process $\mathbf{x}_t$, which is governed by the following Itô stochastic differential equation (SDE):

$$d\mathbf{x}_t = \mathbf{f}(\mathbf{x}_t, t)dt + g(t)d\mathbf{w}, \tag{1}$$

where $t \in [0, 1]$, $\mathbf{f}(\cdot, t) : \mathbb{R}^d \to \mathbb{R}^d$ is the drift coefficient, $g(\cdot) : \mathbb{R} \to \mathbb{R}$ is the diffusion coefficient of $\mathbf{x}_t$, and $\mathbf{w} \in \mathbb{R}^d$ is a standard Wiener process.

The time-reversed version of this diffusion process, derived from the Fokker–Planck equations (Song and Ermon, 2019), is also a diffusion process. This reverse-time SDE is defined as:

$$d\mathbf{x}_t = \left(\mathbf{f}(\mathbf{x}_t, t) - g(t)^2 \nabla_{\mathbf{x}_t} \log p_t(\mathbf{x}_t)\right) dt + g(t)d\bar{\mathbf{w}}, \tag{2}$$

where $\bar{\mathbf{w}}$ is a standard Wiener process. Equation 2 produces the same marginal distributions as the forward diffusion process defined by Equation 1.

Equation 2 must be solved to generate samples from this model. The score-matching approach simplifies this process by redefining $\mathbf{f}(\mathbf{x}_t, t) = h(t)\mathbf{x}_t$, where $h : \mathbb{R} \to \mathbb{R}$ typically assumes an affine form, such as $\mathbf{f}(\mathbf{x}_t, t) = \mathbf{x}_t$. Moreover, $g(\cdot)$, which governs the noise intensity added to the process, often follows a linear or exponential schedule in time.

This reformulation enables a simplification of solving Equation 2 by focusing solely on learning $\nabla_{\mathbf{x}} \log p_t(\mathbf{x}_t)$ for all $t$. With this approach, and initializing with random noise, e.g., $\mathbf{x}_0 \sim \mathcal{N}(0, \mathbf{I})$, the process gradually converges to the data distribution $p_{\text{data}}$, reaching it at $t = 1$.

**(Conditional) Flow Matching (CFM).** Flow-matching (FM) transforms a simple prior distribution into a complex target distribution. FM is defined by a time-dependent vector field $\mathbf{v}_t$, which governs the dynamics of an ordinary differential equation (ODE):

$$\frac{d}{dt}(\mathbf{x}_t) = \mathbf{v}_t(\mathbf{x}_t) \tag{3}$$

The solution to this ODE, represented by $\phi(\mathbf{x}_t)$, evolves along the vector field $\mathbf{v}_t$ from time 0 to time 1, producing $\mathbf{x}_1 \sim p_1$. As samples move along the vector field $\mathbf{v}_t$, $p_t(\mathbf{x}_t)$ changes over time, described by the continuity equation.

$$\frac{\partial p_t}{\partial t} = -\nabla \cdot (p_t \mathbf{v}_t) \tag{4}$$

where $p_t(\mathbf{x}_t)$ is determined by $\mathbf{v}_t(\mathbf{x}_t)$, $\nabla\cdot$ is the divergence operator, with $p_1 \approx p_{\text{data}}$.

If the probability path $p_t(\mathbf{x}_t)$ and the vector field $\mathbf{v}_t(\mathbf{x}_t)$ are given, and if $p_t(\mathbf{x}_t)$ can be efficiently sampled, we can then define a parameterized vector field $\hat{\mathbf{v}}_t(\mathbf{x}_t; \boldsymbol{\theta})$ using a neural network with weights $\boldsymbol{\theta}$. This parameterized vector field approximates $\mathbf{v}_t(\mathbf{x}_t)$, and our goal is to learn how to generate samples from the ODE defined in equation 3. The parameters of the neural network are trained using the following objective: $\mathcal{L}_{\text{FM}}(\boldsymbol{\theta}) := \mathbb{E}_{t \sim \mathcal{U}(0,1), \mathbf{x}_t \sim p_t(\mathbf{x}_t)} \|\hat{\mathbf{v}}_t(\mathbf{x}_t; \boldsymbol{\theta}) - \mathbf{v}_t(\mathbf{x}_t)\|^2$ to solve the ODE defined in equation 3. Here, $\mathbf{v}_t(\mathbf{x}_t)$ can be defined as $\mathbf{v}_t(\mathbf{x}_t) := \mathbb{E}_{q(\mathbf{z})}\left[\frac{\mathbf{v}_t(\mathbf{x}_t|\mathbf{z})p_t(\mathbf{x}_t|\mathbf{z})}{p_t(\mathbf{z})}\right]$ with some conditional variable $\mathbf{z}$ with distribution $q(\mathbf{z})$; refer to Theorem 3.1 in Tong et al. (2023a) for more details.

However, this learning objective becomes intractable for general source and target distributions. To mitigate this, we can focus on cases where the conditional probability paths $p_t(\mathbf{x}_t|\mathbf{x}_1)$ and vector fields $\mathbf{v}_t(\mathbf{x}_t|\mathbf{x}_1)$ associated with $p_t(\mathbf{x}_t)$ and $\mathbf{v}_t(\mathbf{x}_t)$ are known and have simple forms (Tong et al., 2023b). In such cases, we can recover the vector field $\mathbf{v}_t(\mathbf{x}_t)$ using an unbiased stochastic objective known as the Conditional Flow Matching (CFM) loss, defined as:

$$\mathcal{L}_{\text{CFM}}(\boldsymbol{\theta}) := \mathbb{E}_{t, \mathbf{x}_1, \mathbf{x}_t} \|\hat{\mathbf{v}}_t(\mathbf{x}_t|\mathbf{x}_1; \boldsymbol{\theta}) - \mathbf{v}_t(\mathbf{x}_t)\|^2 \tag{5}$$

where $t \sim \mathcal{U}[0, 1]$, $\mathbf{x}_1 \sim p_{\text{data}}$, and $\mathbf{x}_t \sim p_t(\mathbf{x}_t|\mathbf{x}_1)$. The training objective in Equation 5 ensures that the marginalized vector field $\hat{\mathbf{v}}_t(\mathbf{x}_t|\mathbf{x}_1; \boldsymbol{\theta})$, denoted as $\hat{\mathbf{v}}_t(\mathbf{x}_t)$, generates $p_t(\mathbf{x}_t)$, similar to FM models.

**CFMs vs Diffusion Models.** CFMs share similarities with diffusion models; both rely on defining continuous probability paths over time. However, diffusion models use stochastic processes to transform data distributions, typically governed by an SDE. However, CFMs utilize a vector field to directly map an initial prior distribution to the target distribution via a deterministic process (Equation 4). Diffusion models, such as denoising score matching models, rely on stochastic diffusion paths to approximate the data distribution. CFMs bypass this stochastic process by constructing a conditional vector field and sampling from the ODE defined in Equation 3.

**Optimal Transport CFMs (OT-CFMs).** The CFM objective defined in Equation 5 appears simple. However, it is nearly intractable due to the absence of prior knowledge regarding an appropriate mapping that links between the initial ($p_0$) and target ($p_{\text{data}}$) distributions. To address this challenge, optimal transport (OT) (Tong et al., 2020) is used to couple the initial and target distributions. This approach seeks a mapping from $p_0$ to $p_1$ that minimizes the displacement cost between these two distributions, resulting in flows that can be integrated accurately. This CFM variant is known as Optimal Transport CFMs (OT-CFMs), and the conditional vector field corresponds to this is

$\hat{\mathbf{v}}_t(\mathbf{x}_t|\mathbf{x}_1) = \frac{\mathbf{x}_1 - \mathbf{x}_t}{1-t}$. While OT improves training efficiency in CFMs, traditional sampling methods for CFMs require over a hundred function evaluations (NFE) to produce high-quality samples (Tong et al., 2023a). The lack of flexibility in sampling steps during training further limits their ability to efficiently generate samples, often resulting in a high NFE.

Additionally, the OT-CFM vector field, which can be reformulated as $\hat{\mathbf{v}}_t(\mathbf{x}_t|\mathbf{x}_1) = \mathbf{x}_1 - \mathbf{x}_0$ Pooladian et al. (2023), focuses on exactly transporting all points $\mathbf{x}_0$ to $\mathbf{x}_1$ along straight-line paths. This implies that all points along the path from $0$ to $t$ means that the transport map shares the same regression weights when training with the $\mathcal{L}_{\text{CFM}}(\boldsymbol{\theta})$ objective. However, this uniform weighting across the trajectory may overlook that points closer to $t = 1$ need finer attention to detail and, consequently, a more complex flow; this may limit the model's ability to effectively represent complex structures in the final generated samples. Furthermore, the deterministic sample generation in CFMs, $\mathbf{x}_t = \mathbf{x}_{t-\Delta t} + \Delta t \hat{\mathbf{v}}_t(\mathbf{x}_t; \boldsymbol{\theta})$ (see Figure 2.B for visualization), may restrict the flexibility of CFM, as it prevents the introduction of stochasticity that could otherwise enhance the diversity and richness of generated samples. This might manifest in a lack of expressiveness when generating fine-grained or high-dimensional data. These limitations underscore the need for new approaches to sample diversity and NFE of CFMs. Our paper tackles these limitations by proposing an alternative vector field that better balances deterministic transport with elements of stochasticity to improve sample quality.

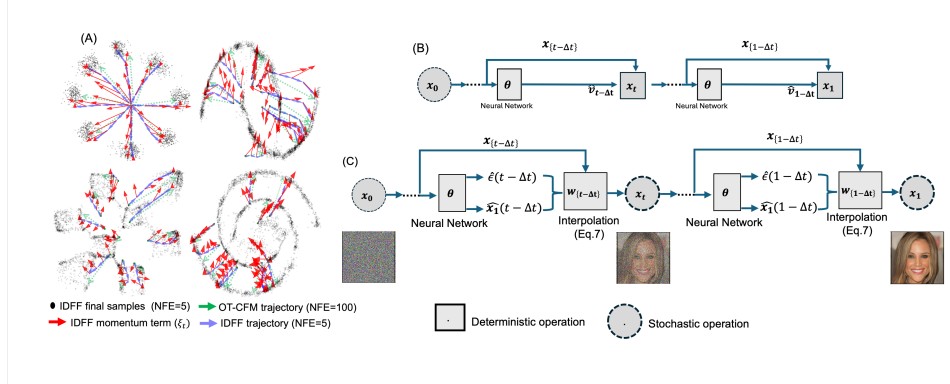

Figure 2: A) Comparison of trajectory sampling between IDFF and OT-CFMs: The figure displays 4096 final samples generated by IDFF. As shown, IDFF takes larger steps toward the target distribution, guided by the momentum term. While OT-CFM follows a nearly straight path to reach the target distribution, it requires a higher number of function evaluations (NFEs). B) OT-CFMs sampling process. C) IDFF sampling process. In this process, $\hat{\mathbf{x}}_1(.)$ approximates the data sample $\mathbf{x}_1$, $\hat{\epsilon}(.)$ approximates the scores associated with $\boldsymbol{\xi}_t$, and $\mathbf{w}_t(.)$ is the calculated vector field by equation 7. For further details, see Algorithm 2. The key difference between IDFF and OT-CFMs is that instead of directly generating the vector field with, IDFF generates $\hat{\mathbf{x}}_1(.)$ and $\hat{\epsilon}(.)$, and then reconstructs the vector field, following it. Additionally, IDFF uses a Gaussian distribution to sample $\mathbf{x}_t$ rather than calculating it deterministically.

**Hamiltonian Monte Carlo (HMC).** HMC is a powerful sampling algorithm designed to improve efficiency in exploring complex probability distributions by introducing auxiliary momentum variables. In HMC, particle dynamics are simulated in a potential field using Hamiltonian dynamics, which describe the total energy of a system through the Hamiltonian function $H(\mathbf{x}, \mathbf{p}) = U(\mathbf{x}) + K(\mathbf{p})$. $\mathbf{x}$ represents the position, $\mathbf{p}$ represents momentum, $U(\mathbf{x})$ represents the potential energy related to the target distribution, and $K(\mathbf{p}) = \frac{1}{2}\mathbf{p}^T\mathbf{M}^{-1}\mathbf{p}$ is the kinetic energy, with $\mathbf{M}$ being the mass matrix. The dynamics of this system evolve according to:

$$\frac{d\mathbf{x}}{dt} = \nabla_{\mathbf{p}}H(\mathbf{x}, \mathbf{p}) = \mathbf{M}^{-1}\mathbf{p}, \quad \frac{d\mathbf{p}}{dt} = -\nabla_{\mathbf{x}}H(\mathbf{x}, \mathbf{p}) = -\nabla U(\mathbf{x}).$$

These dynamics preserve the system's total energy, enabling efficient exploration of the probability space. HMC achieves this by leveraging momentum to facilitate directed movement, reducing random walk behavior, and overcoming local energy barriers. These properties lead to faster convergence to

216
217
218
the target distribution with fewer function evaluations. HMC generates a sample from some target distributions by following Hamiltonian dynamics in an extended state space $(\mathbf{x}, \mathbf{p})$. IDFF attempts to design a flow-matching procedure with a similar intuition.

219
220
221

## 3 Implicit Dynamical Flow Fusion Model (IDFF)

222
223
224
225
226
IDFF aims to improve sampling efficiency while enhancing sample quality of OT-CFMs through two key innovations: (1) the incorporation of a learnable Hamiltonian momentum term, $\boldsymbol{\xi}_t$, into the conditional vector field, which enhances sampling efficiency; and (2) the proposal of a new objective function that learns the IDFF's vector field in input space.

227
228

### 3.1 Learning CFMs with Momentum

229
230
231
To improve sampling efficiency while preserving distribution accuracy in CFMs, we introduce a learnable Hamiltonian momentum term, $\boldsymbol{\xi}_t$, integrated with the original vector field $\mathbf{v}_t(\mathbf{x}_t)$. This integration yields a new, more flexible vector field:

232
233
$$\tilde{\mathbf{v}}_t(\mathbf{x}_t) = \sqrt{1 - \sigma_t^2}\mathbf{v}_t(\mathbf{x}_t) + \sigma_t^2\boldsymbol{\xi}_t, \tag{6}$$

234
where $\sigma_t$ is a time-dependent interpolation factor balancing $\mathbf{v}_t$ and $\boldsymbol{\xi}_t$.

235
236
237
238
239
240
241
*Interpolating the vector field:* We define the interpolation factor $\sigma_t$ as $\sigma_t = \sigma_0\sqrt{t(1-t)}$, where $\sigma_0$ is a hyperparameter that controls the overall influence of the momentum term. While the IDFF vector field converges to OT-CFMs in either limit, outside of it, it enables the flow to leverage the momentum variables to guide the flow based on the momentum variables. Our empirical results demonstrate that this approach significantly improves sampling efficiency by substantially reducing the NFEs required. Our approach also overcomes the limitations of uniform weighting across trajectories by prioritizing samples near $t = 0, 1$, allowing finer attention to complex structures; see Figure 2.A.

242
243
244
245
246
247
248
249
250
251
252
253
254
255
256
*Designing momentum variables:* The term $\boldsymbol{\xi}_t$ is designed to guide the new vector field toward the correct probability flow for the next time step, ensuring its role in accurately modeling the distribution. There are many potential choices to make here. Still, we take inspiration from Hamiltonian Monte Carlo, which showcases how to leverage momenta to sample complex multi-modal posterior distributions rapidly Neal (2012b). A potential design for $\boldsymbol{\xi}_t$ is $\boldsymbol{\xi}_t = \gamma\nabla_x \log p_t(\mathbf{x}_t)$, which aligns with the gradient of the log probability of the samples $\mathbf{x}_t$. If $p_t(\mathbf{x}_t \mid \mathbf{x}_1)$ is a Gaussian with variance $\sigma_t^2$, $\boldsymbol{\xi}_t$ can be simplified to $-\frac{\boldsymbol{\epsilon_0}}{\sigma_t}$, where $\boldsymbol{\epsilon_0} \sim \mathcal{N}(\mathbf{0}, \mathbf{I})$. $\boldsymbol{\xi}_t$ directs the flow similarly to HMC's momentum, promoting efficient exploration of complex distributions. The theoretical connection between HMC and IDFF can be understood through their respective probability flows. HMC generates samples by following Hamiltonian dynamics in an extended state space $(\mathbf{x}, \mathbf{p})$. In contrast, IDFF's probability flow operates directly in the sample space while incorporating momentum-like behavior through $\boldsymbol{\xi}_t$. Both methods use auxiliary variables to enhance sampling efficiency and preserve the desired target distribution. Alternatively, higher-order choices for momentum terms could capture even more complex interactions in the flow. $\boldsymbol{\xi}_t = \gamma\nabla_x \log p_t(\mathbf{x}_t)$ drives the vector field toward regions of high probability, thereby improving the accuracy of the generated samples. $\gamma$ serves as a weighting factor that controls the influence of this term.

257
258
259
260
To maintain the same probability path as the original CFM, the vector field must satisfy the continuity constraint defined in equation 4. By incorporating $\tilde{\mathbf{v}}_t(\mathbf{x}_t)$ into the continuity equation 4 (detailed in Appendix A.1.1), we derive the IDFF probability flow ODE:

261
262
$$\mathbf{w}_t(\mathbf{x}_t) = \tilde{\mathbf{v}}_t(\mathbf{x}_t) - \frac{\sigma_t^2}{2}\nabla \log p_t(\mathbf{x}_t) = \sqrt{1 - \sigma_t^2}\mathbf{v}_t(\mathbf{x}_t) + \frac{(2\gamma - 1)}{2}\sigma_t^2\nabla \log p_t(\mathbf{x}_t), \tag{7}$$

263
264
265
To ensure that our new marginal vector field $\mathbf{w}_t(\mathbf{x}_t)$ converges to the standard $\mathbf{v}_t(\mathbf{x}_t)$ at the start and end points of the process, $\sigma_t$ must approach zero as $t \to 0, 1$. This convergence is crucial for maintaining consistent marginal distributions at these points.

266
267
268
269
Therefore, $\mathbf{w}_t(\mathbf{x}_t)$ now replaces the vector field $\mathbf{v}(\mathbf{x}_t)$ in the probability flow ODE (equation 3), enabling the construction of a new generative process that follows the same probability path $p_t(\mathbf{x}_t)$. This process starts with $\mathbf{x}_0 \sim \mathcal{N}(0, \mathbf{I})$ and follows $p_t(\mathbf{x}_t)$ to produce $\mathbf{x}_1$. The resulting model thus *implicitly* models the score function using flow matching which motivates the name *Implicit Diffusion Flow Fusion*.

**Lemma 1.** *Given that* $\mathbf{v}_t(\mathbf{x}_t)$ *generates* $p_t(\mathbf{x}_t)$ *and* $\sigma_t$ *approaches 0 for* $t \in 0, 1$, *the IDFF probability flow ODE (Eq. 7) follows the same marginal distribution as the CFMs with the ODE defined in equation 3.*

*Proof Schema for Lemma 1.* Our goal is to show that the IDFF probability flow ODE shares the same marginal distribution as CFMs when $\sigma_t$ is small for $t \in 0, 1$.

- **Step 1: Define the vector fields.** We introduce $\tilde{\mathbf{v}}_t(\mathbf{x}_t)$ governed by the $\boldsymbol{\xi}_t$ and $\mathbf{v}_t(\mathbf{x}_t)$ from the original CFM.

- **Step 2: Apply the continuity equation.** By substituting $\tilde{\mathbf{v}}_t(\mathbf{x}_t)$ into the continuity equation, we derive the IDFF ODE, $\mathbf{w}_t(\mathbf{x}_t)$.

- **Step 3: Analyze boundary conditions.** We show that at $t = 0$ and $t = 1$, where $\sigma_t$ approaches 0, the IDFF ODE converges to the CFM-ODE, ensuring the same marginal distributions.

$\square$

Appendix A.1.1 provides the complete proof. This lemma and its proof establish the consistency of IDFF concerning existing CFMs. The approach showcases how to design a new flow augmented with momentum variables with an interpolant that guarantees that it matches OT-CFM in the limit. In what follows, we discuss how to parameterize and learn such a flow.

## 3.2 Learning Flows in Input Space

One of the challenges in our formalism is learning $\mathbf{w}_t$. This challenge arises because $\mathbf{v}_t$ operates in vector space while $\boldsymbol{\xi}_t$ remains in the input space. Rather than training one part of the model in vector space and the other in input space, we define a new objective defined entirely within the input space to address this. Learning the vector field in the input space implies optimizing the vector itself, which is the reasoning behind naming the IDFF model an implicit model of dynamical flows. We begin by defining an OT path $p_t(\mathbf{x}_t|\mathbf{x}_1)$, which acts as a probability bridge between $\mathbf{x}_0$ and $\mathbf{x}_1$. This path is a stochastic interpolator between the source $\mathbf{x}_1$ and target $\mathbf{x}_0$, as follows:

$$p_t(\mathbf{x}_t|\mathbf{x}_1) = \mathcal{N}(\mathbf{x}_t \mid t\mathbf{x}_1 + (1-t)\mathbf{x}_0, \sigma_t^2\mathbf{I}), \ \ \sigma_t = \sigma_0\sqrt{t(1-t)} \tag{8}$$

The corresponding conditional vector field is given by:

$$\hat{\mathbf{v}}_t(\mathbf{x}_t|\mathbf{x}_1) = \frac{\mathbf{x}_1 - \mathbf{x}_t}{1-t} \tag{9}$$

This definition aligns with the conditional OT paths between $\mathbf{x}_1$ and $\mathbf{x}_0$. For large datasets, computing this OT map can be challenging. We employ a minibatch approximation of OT to address this, similar to (Fatras et al., 2021). While this introduces some errors compared to the exact OT solution, it has proven effective in numerous applications such as domain adaptation and generative modeling (Genevay et al., 2018).

Current CFM objectives typically use a loss function (defined in equation 5) that optimizes the prediction of the vector field. We propose an alternative approach: defining a denoising neural network $\hat{\mathbf{x}}_1(\mathbf{x}_t, t; \theta)$ and reparameterizing $\hat{\mathbf{v}}_t(\mathbf{x}_t \mid \mathbf{x}_1; \theta)$ as follows:

$$\hat{\mathbf{v}}_t(\mathbf{x}_t \mid \mathbf{x}_1; \theta) = \frac{\hat{\mathbf{x}}_1(\mathbf{x}_t, t; \theta) - \mathbf{x}_t}{1-t} \tag{10}$$

Since both $\hat{\mathbf{x}}_1(\mathbf{x}_t, t; \theta)$ and $p_t(\mathbf{x}_t|\mathbf{x}_1)$ are easy to sample, we use a single neural network with two separate output heads: the first head, $\hat{\mathbf{x}}_1(\mathbf{x}_t, t; \theta)$, to approximate the data samples, and the second, $\hat{\boldsymbol{\epsilon}}(\mathbf{x}_t, t; \theta)$, to approximate the score function. The network generates these outputs simultaneously. The score function $\nabla_{\mathbf{x}_t} \log p_t(\mathbf{x}_t \mid \mathbf{x}_1)$, using a Gaussian $p_t(\mathbf{x}_t \mid \mathbf{x}_1)$, can be simplified to $-\frac{\boldsymbol{\epsilon}_0}{\sigma_t}$, where $\boldsymbol{\epsilon}_0 \sim \mathcal{N}(\mathbf{0}, \mathbf{I})$. The neural network can be trained using the following loss function:

$$\mathcal{L}_{\text{IDFF}}(\theta) = \mathbb{E}_{t, \mathbf{x}_0, \mathbf{x}_1, \mathbf{x}_t} \left[ \beta(t)^2 |\hat{\mathbf{x}}_1(\mathbf{x}_t, t; \theta) - \mathbf{x}_1|^2 + \lambda(t)^2 |\hat{\boldsymbol{\epsilon}}(\mathbf{x}_t, t; \theta) - \nabla_x \log p_t(\mathbf{x}_t|\mathbf{x}_1)|^2 \right] \tag{11}$$

Where $t \sim \mathcal{U}(0,1), \mathbf{x}_0 \sim \mathcal{N}(\mathbf{0}, \mathbf{I}), \mathbf{x}_1 \sim p_{\text{data}}, \mathbf{x}_t \sim p_t(\mathbf{x}_t|\mathbf{x}_1)$. Here $\lambda(\cdot), \beta(\cdot)$ are two sets of positive weights.

Our proposed loss function comprises two terms: The first term optimizes the denoising model $\hat{\mathbf{x}}_1(\mathbf{x}_t, t; \theta)$, with $\mathbf{x}_t$ as the noisy input. The second term represents a scoring model that approximates $\nabla_x \log p_t(\mathbf{x}_t|\mathbf{x}_1)$. The first term of the loss incorporates a $\beta(t)$ weighting schedule, which emphasizes the quality of samples as they approach $t = 1$. For more details about the loss function, please refer to Appendix A.1.2. The complete training process is described in Algorithm 1.

**Sampling**    After training $\hat{\mathbf{x}}_1(.; \theta)$ and $\boldsymbol{\epsilon}(.; \theta)$, we can generate samples from IDFF by solving an SDE starting from the source samples $\mathbf{x}_0$. Refer to Figure 2.C for visualization. This procedure is outlined in Algorithm 2. The SDE is defined as $d\mathbf{x}_t = \mathbf{w}_t dt + \sigma_t d\mathbf{w}$, with initial samples $\mathbf{x}_0$ and target samples $\mathbf{x}_1$ at $t = 1$. Therefore, we can draw samples from $p(\mathbf{x}_0)$, run them through the SDE, and obtain the final samples $\mathbf{x}_1$. First, we draw $\mathbf{x}_0 \sim \mathcal{N}(0, \mathbf{I})$, then compute $\hat{\mathbf{x}}_1(\mathbf{x}_0, 0, k; \theta)$. Next, we calculate $\hat{\mathbf{v}}_t$ using equation 10, followed by the computation of $\mathbf{w}_t$. After calculating $\mathbf{w}_t$ and using the SDE, we compute the mean for the next time step as $\boldsymbol{\mu}_{t+\Delta t} = (\mathbf{x}_t + \mathbf{w}_t \Delta t)$, and finally draw a sample from a small neighborhood of it: $\mathbf{x}_{t+\Delta t} \sim \mathcal{N}(\mathbf{x}_t + \mathbf{w}_t \Delta t, \sigma_t^2 \Delta t \mathbf{I})$. The $\Delta t$ is determined by the NFE as $\Delta t = \frac{1}{\text{NFE}}$, where smaller NFE results in finer time discretization and tighter neighborhoods.

**Likelihood calculation.**    To evaluate the likelihood of data under IDFFs, we can leverage the change in the probability density as the sample $\mathbf{x}_t$ evolves according to the velocity field $\mathbf{w}_t$ as described by the continuity equation 4, which gives the time evolution of the probability density under the learned flow Lipman et al. (2022):

$$\log p_1(\mathbf{x}_1) = \log p_0(\mathbf{x}_0) - \int_0^1 \nabla \cdot \mathbf{w}_t(\mathbf{x}_t) \, dt \tag{12}$$

This integral is numerically approximated, often using Monte Carlo methods.

### 3.3   IDFF TIME-SERIES ADAPTATION

To use IDFF for time-series applications, we need to modify the training and sampling algorithms for IDFF as shown in Algorithms 4 and 3, respectively. To accomplish that, we define another random variable, $k$, which is a discrete variable and can take values of $1, ..., K$ with equal probability, where $K$ is the length of a time series. Therefore, $k$ represents the index of the data sample in a time series. We also have $t$ as a continuous time variable interpolated between two subsequent values of $k$ (i.e., $k$ and $k - 1$). Therefore we have $\forall k \in 1, ..., K \rightarrow p(t \mid k) \sim \mathcal{U}(0, 1)$. This conditioning does not change the original assumption over $t$ since $k$ can take any values up to the time-series length with the same probability. With this in hand, we can directly pass the $k$ as additional inputs to $\hat{\boldsymbol{\epsilon}}_1(\mathbf{x}_t, t, k; \theta)$ and $\hat{\mathbf{x}}_1(\mathbf{x}_t, t, k; \theta)$ and suggest Algorithm 1 for training and Algorithm 2 for sampling time-series data with IDFF. Note that for static data, the sequence length ($K$) equals one; otherwise, it equals the time series length.

## 4   RELATED WORK

Recent advancements in CFMs include Schrödinger bridge-based methods (Richter et al., 2023) and conjugate/splitting-based integrators (Pandey et al., 2023), both aimed at improving sampling efficiency. However, these methods still struggle to significantly reduce the NFEs, which remains a critical factor for computational efficiency.

In diffusion models, denoising diffusion implicit models (DDIMs) (Song et al., 2020b) were introduced to address the issue of large NFEs by offering a more efficient sampling process. DDIMs deterministically generate samples from latent variables, unlike DDPMs, which rely on Langevin dynamics. By employing a variational approach, DDIMs accelerate sampling while preserving output quality. Other approaches to enhance sampling in diffusion-based generative models include distillation frameworks such as Flash (Kohler et al., 2024), which generate diverse samples in fewer steps by mitigating training-inference discrepancies.

A distinguishing feature of IDFF is its ability to jointly learn an implicit flow from the generated samples alongside the scoring model. This sets it apart from models like DDIM, which focuses solely

**Algorithm 1** IDFF training algorithm.

**Input:** dataset distribution $p_1(\mathbf{x}_1)$, initial distribution $p_0(\mathbf{x}_0)$, bandwidth $\sigma_0$, a set of positive weights $\lambda(.)$ and $\beta(.)$, initialized networks $\hat{\mathbf{x}}_1(.;\theta)$ and $\hat{\boldsymbol{\epsilon}}(.;\theta)$,

**while** Training **do**
    $\mathbf{x}_1 \sim p_1(\mathbf{x}_1),\ \mathbf{x}_0 \sim p_0(\mathbf{x}_0)$
    $\pi \leftarrow \text{OT}(\mathbf{x}_1, \mathbf{x}_0)$
    $(\mathbf{x}_0, \mathbf{x}_1) \sim \pi$
    $t \sim \mathcal{U}(0, 1)$
    $\boldsymbol{\mu}_t \leftarrow t\mathbf{x}_1 + (1-t)\mathbf{x}_0$
    $\sigma_t \leftarrow \sigma_0 \sqrt{t(1-t)}$
    $\mathbf{x}_t \sim \mathcal{N}(\boldsymbol{\mu}_t, \sigma_t^2 \mathbf{I})$
    $\mathcal{L}_{\text{IDFF}}(\theta) \leftarrow \beta(t)^2 \|\hat{\mathbf{x}}_1(\mathbf{x}_t, t, k; \theta) - \mathbf{x}_1\|^2 + \lambda(t)^2 \|\boldsymbol{\epsilon}(\mathbf{x}_t, t; \theta) - \nabla_x \log p_t(\mathbf{x}_t | \mathbf{x}_1)\|^2$
    $\theta \leftarrow \text{Update}(\theta, \nabla_\theta L(\theta))$
**end while**
**return** $\{\hat{\mathbf{x}}_1(.;\theta), \hat{\epsilon}(.;\theta)\}$

**Algorithm 2** IDFF Sampling algorithm

**Input:** $\hat{\mathbf{x}}_1(.;\theta)$ and $\hat{\boldsymbol{\epsilon}}(.;\theta)$, bandwidth $\sigma_0$, time step size $\Delta t$, and $\gamma$.
$\mathbf{x}_0^0 \sim \mathcal{N}(0, \mathbf{I})$
**for** $t$ in $[0, 1/\Delta t]$ **do**
    $\sigma_t \leftarrow \sigma_0 \sqrt{t(1-t)}$
    $\mathbf{w}_t \leftarrow \sqrt{(1-\sigma_t^2)} \frac{\hat{\mathbf{x}}_1(\mathbf{x}_t, t; \theta) - \mathbf{x}_t}{1-t} + \frac{2\gamma - 1}{2} \sigma_t \hat{\epsilon}(\mathbf{x}_t, t; \theta)$
    $\mathbf{x}_{t+\Delta t} \sim \mathcal{N}(\mathbf{x}_t + \mathbf{w}_t \Delta t, \sigma_t^2 \Delta t \mathbf{I})$
**end for**
**return** Samples $\{\mathbf{x}_1\}$

on learning a scoring function. By leveraging OT to couple samples between $t = 0$ and $t = 1$, IDFF accelerates convergence during training while maintaining a low NFE and improving sample quality.

Our image generation experiments, presented in Tables 1 and 2, demonstrate IDFF's superior performance. Compared to $[SF]^2M$ (Tong et al., 2023b), OT-CFM(Pooladian et al., 2023), DDPM (Ho et al., 2020), DDIM (Song et al., 2020b), EDM (Karras et al., 2022), and DPM-Solver (Lu et al., 2022a) (designed to reduce NFEs in diffusion models), IDFF consistently outperforms these models in both efficiency and quality.

In summary, IDFF combines conditional flow and score-based learning, providing an efficient framework for generative modeling that reduces computational costs while achieving high-quality samples in training iterations, much fewer than diffusion-based models and comparable to CFM (see Appendix A.3 for details). The next section elaborates on these comparisons.

Table 1: Comparison of Likelihood (BPD), FID, and NFE between IDFF and various methods on the CIFAR-10 dataset. Additional results are provided in Table 7

| Model | NLL↓ | FID↓ | NFE↓ |
|---|---|---|---|
| DDPM | 3.12 | 7.48 | 274 |
| EDM | – | 16.57 | 10 |
| DDIM | – | 6.84 | 20 |
| DDIM | – | 13.36 | 10 |
| DPM-Solver | – | 6.03 | 12 |
| Score Matching | 3.16 | 19.94 | 242 |
| ScoreFlow | 3.09 | 20.78 | 428 |
| OT-CFM | **2.99** | 6.35 | 142 |
| OT-CFM | – | 11.87 | 10 |
| $[SF]^2$M | – | 10.13 | 10 |
| FM | – | 14.36 | 10 |
| IC-CFM | – | 13.68 | 10 |
| IDFF (Ours) | 3.09 | **5.87** | **10** |

## 5 EXPERIMENTS

We conducted two general classes of experiments to demonstrate the effectiveness of IDFF in generative modeling on static data, such as images and time-series data, including simulated chaotic systems (e.g., attractors), molecular dynamics, and sea surface temperature forecasting. The anonymous code for our implementation is accessible at Anonymous Repository.

### 5.1 IMAGE GENERATION

*IDFFs exhibit superior performance compared to CFMs in image generation.* We trained IDFF models on datasets such as CIFAR-10 and CelebA (additional results for ImageNet-64, CelebA-HQ, LSUN Bedrooms, and LSUN Church are available in Appendix A.6). We utilized a CNN-based UNet (Dhariwal and Nichol, 2021) to simultaneously model both $\hat{\mathbf{x}}_1(., t; \theta)$ and $\boldsymbol{\epsilon}(., t; \theta)$ from Equation 11 for all image data except ImageNet-64. For ImageNet-64, we employed a DiT architecture ('DiT-L/4')(Peebles

Table 2: Comparison of FID and NFE metrics between IDFF and various methods on the CelebA ($64 \times 64$) dataset.

| Model | FID↓ | NFE↓ |
|---|---|---|
| DDPM | 45.20 | 100 |
| DDIM | 13.73 | 20 |
| DDIM | 17.33 | 10 |
| FastDPM | 12.83 | 50 |
| IDFF (Ours) | **11.83** | **10** |

and Xie, 2022). To implement this, we doubled the input channels of the networks and fed the augmented input, $(\mathbf{x}_t, \mathbf{x}_t)$, into the network. The outputs were then split into $(\hat{\mathbf{x}}_1(., t; \theta), \boldsymbol{\epsilon}(., t; \theta))$.

To ensure a fair comparison with other models, we used commonly employed metrics to evaluate CFMs' performance, including negative log-likelihood (NLL) using equation 12 measured in bits per dimension (BPD) Lipman et al. (2022), sample quality as measured by the Frechet Inception Distance (FID), and the average NFE required for the reported FID and NLL, averaged over 50k samples, to assess the computational efficiency of our method. We generate 50k images to calculate the FID against the corresponding reference statistics of each dataset. All models use the same architecture, with specific details provided in Appendix A.5.

We also examined the effect of $\sigma_0$ on sample quality for both CIFAR-10 and CelebA datasets in Appendix A.2, finding that a value of $\sigma_0 = 0.6$ achieved the best FID for both datasets, as shown in Table 1 and Table 2. IDFF achieves comparable FID scores with ten times fewer NFEs than CFM and DDPM on the CIFAR-10 and CelebA datasets. IDFF's qualitative results are provided in Figure 1 (additional samples are available in Appendix A.6).

Figure 4 (A) displays the FID score for the initial 50k generated samples plotted against NFEs for both the CIFAR-10 and CelebA datasets. Sample quality improves rapidly when the NFE exceeds 4, highlighting the significantly lower sampling cost of IDFF compared to traditional CFMs and diffusion-based models.

## 5.2 TIME SERIES GENERATION

To assess the performance of IDFF in modeling time series data, we evaluated IDFF on three distinct tasks: 3D-attractor generation (see Appendix A.7), molecular dynamics simulation, and sea surface temperature forecasting.

### 5.2.1 MOLECULAR DYNAMICS SIMULATION

*IDFFs can predict dynamics for complex combinatorial structures such as molecules.* Molecular dynamics simulation is a crucial tool in quantum mechanics for understanding the dynamics of molecular behavior at the atomic level (Cazorla and Boronat, 2017). Here, we simulated a fully extended polyalanine structure for 400 picoseconds in a vacuum environment at a temperature of $T = 300K$. This molecular structure consists of 253 atoms with 46 dihedral angles. We trained an IDFF model to accurately generate the dihedral angles for the polyalanine structure from scratch. The distributions of the actual and generated dihedral angles for all 46

Table 3: MAE, RMSE, and CC between true and predicted trajectories for the MD simulation.

| Method | MAE↓ | RMSE ↓ | CC (%)↑ |
|---|---|---|---|
| SRNN | $82.6_{\pm 28}$ | $91.9_{\pm 25}$ | $10.2_{\pm 0.27}$ |
| DVAE | $78.1_{\pm 27}$ | $88.1_{\pm 25}$ | $30.4_{\pm 0.35}$ |
| NODE | $25.3_{\pm 6.3}$ | $28.8_{\pm 6.2}$ | $10.5_{\pm 0.41}$ |
| OT-CFM | | | |
| NFE=50 | $13.3_{\pm 1.1}$ | $16.3_{\pm 2.4}$ | $86.1_{\pm 0.1}$ |
| NFE=10 | $15.5_{\pm 1.3}$ | $19.2_{\pm 2.7}$ | $82.4_{\pm 0.2}$ |
| IDFF | | | |
| NFE=5 | $\mathbf{10.3}_{\pm 1.3}$ | $\mathbf{14.9}_{\pm 2.7}$ | $\mathbf{93.2}_{\pm 0.05}$ |
| NFE=12 | $\mathbf{9.5}_{\pm 1.1}$ | $\mathbf{13.4}_{\pm 2.8}$ | $\mathbf{95.2}_{\pm 0.1}$ |

angles are presented in Figure 3 (A) and (B), respectively. Figure 3 (C) illustrates the dihedral angles for one of the alanine molecules and its associated dihedral angles. A complete trajectory showcasing a pair of actual and generated dihedral angles is depicted in Figure 3 (D).

To assess the performance of IDFF in the MD simulation, we utilized several evaluation metrics, including root mean squared error (RMSE), mean absolute error (MAE), and correlation coefficients (CC) between the generated and actual trajectories of dihedral angles. We benchmarked IDFF against well-known dynamical models for time series, including dynamical VAE models such as SRNNs (Fraccaro et al., 2016), VRNNs (Chung et al., 2015), and Neural ODEs (NODEs) (Garnelo et al., 2018). As shown in Table 3, IDFF outperforms the baselines by a large margin, indicating its potential for simulating molecular dynamics from scratch.

### 5.2.2 SEA SURFACE TEMPERATURE FORECASTING

*IDFFs can predict spatiotemporal dynamics for sea-surface temperature.* This experiment centers on forecasting sea surface temperature (SST) (Worsfold et al., 2024). Using the NOAA OISSTv2 dataset, which contains daily high-resolution SST images spanning from 1982 to 2021 (Cachay et al.,

2023), we partitioned the data to predict SST intervals from 1 to 7 days ahead. To achieve this, we strategically subsampled eleven grid tiles of size $60 \times 60$ (latitude $\times$ longitude) from the eastern tropical Pacific region (Huang et al., 2021).

We used a class-conditional UNet for the SST experiment, with the class encoder part encoding the long-range time dependencies (i.e., for encoding the days) to allow continuous forecasting using the UNet. See Appendix A.5 for the experiment's network structure and hyperparameter settings. We used the training 4 and sampling 3 algorithms suggested for time-series data. We evaluated the performance of IDFF using the best validation continuous ranked probability score (CRPS) (Matheson and Winkler, 1976) and mean squared error (MSE) on forecasts up to 7 days. CRPS is a proper scoring rule and a popular metric in probabilistic forecasting literature (Gneiting and Katzfuss, 2014). We calculate the CRPS by generating a 20-member ensemble and the MSE on the ensemble mean predictions in Table 4.

Figure 3: (A) True and (B) generated dihedral angles. (C) The dihedral angles for an alanine molecule. (D) True and generated dihedral angle trajectories using IDFF.

To assess IDFF's performance, we compared it against various baseline approaches, including DDPM, DDPM with enabled dropout (Dropout) (Gal and Ghahramani, 2016), DDPM with random perturbation for the initial values (Perturbation) (Pathak et al., 2022), MCVD (Voleti et al., 2022), Dyffusion (Cachay et al., 2023), and OT-CFM (with $NFE = 40$) as shown in Table 4. The results for all the baselines are with $NFE = 1000$. These baselines provide a robust framework for evaluating the IDFF framework in time-series forecasting. The results show that IDFF (with $NFE = 5$) outperforms the baselines in SST forecasting across both metrics, demonstrating great potential for weather forecasting applications. See samples of the forecasting results in Figure 14.

## 6 Conclusion and Discussion

IDFF models represent a significant advancement in generative modeling, particularly for time series data. By offering superior computational efficiency compared to traditional flow models like CFMs, IDFF achieves a favorable balance between computational cost and sample quality. This makes IDFF a compelling choice for tasks requiring efficient and high-quality sample generation. There are myriad future possibilities for the application of IDFF in domains such as music generation (Briot et al., 2017), speech synthesis (Yu and Deng, 2016), and modeling biological time-series data (Anumanchipalli et al., 2019; Golshan et al., 2020; Rezaei et al., 2021; 2023; Gracco et al., 2005).

Table 4: Results for sea surface temperature forecasting of 1 to 7 days ahead, averaged over the evaluation horizon.

| Method | CRPS↓ | MSE↓ |
|---|---|---|
| Perturb. | $0.281 \pm 0.004$ | $0.180 \pm 0.011$ |
| Dropout | $0.267 \pm 0.003$ | $0.164 \pm 0.004$ |
| DDPM | $0.246 \pm 0.005$ | $0.177 \pm 0.005$ |
| MCVD | $0.216$ | $0.161$ |
| Dyffusion | $0.224 \pm 0.001$ | $0.173 \pm 0.001$ |
| OT-CFM | $0.231 \pm 0.005$ | $0.175 \pm 0.006$ |
| IDFF (Ours) | $\mathbf{0.180 \pm 0.024}$ | $\mathbf{0.105 \pm 0.029}$ |

*Limitations and Future Work:* Computing the OT map for large datasets can be challenging. To address this, we use a minibatch approximation of OT; however, increasing the batch size results in longer training times for larger batches. Second, our experiments primarily utilized a CNN-based UNet structure (Dhariwal and Nichol, 2021) as the backbone of the models. We anticipate that some of these limitations may be mitigated by utilizing new neural network structures (Peebles and Xie, 2023).

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

## A APPENDIX

CONTENTS

## A.1 PROOFS

### A.1.1 CONTINUITY PROOF FOR LEMMA 1

We define a vector field as follows:

$$\tilde{\mathbf{v}}_t(\mathbf{x}_t) = \sqrt{1 - \sigma_t^2}\mathbf{v}_t(\mathbf{x}_t) + \sigma_t^2\boldsymbol{\xi}_t, \quad \boldsymbol{\xi}_t = \gamma\nabla\log p_t(\mathbf{x}_t) \tag{13}$$

where $\gamma$ is a coefficient controlling the contribution of the momentum term $\boldsymbol{\xi}_t$.

Consider a Stochastic Differential Equation (SDE) of the standard form:

$$d\mathbf{x}_t = \tilde{\mathbf{v}}_t(\mathbf{x}_t)\,dt + \sigma_t\,d\mathbf{w} \tag{14}$$

with time parameter $t$, drift $\tilde{\mathbf{v}}_t$, diffusion coefficient $\sigma_t$, and $d\mathbf{w}$ representing the Wiener process.

The solution $\mathbf{x}_t$ to the SDE is a stochastic process with probability density $p_t(\mathbf{x}_t)$, characterized by the Fokker-Planck equation:

$$\frac{\partial p_t(\mathbf{x}_t)}{\partial t} = -\nabla \cdot (\tilde{\mathbf{v}}_t(\mathbf{x}_t)p_t(\mathbf{x}_t)) + \frac{\sigma_t^2}{2}\Delta p_t(\mathbf{x}_t) \tag{15}$$

where $\Delta$ represents the Laplace operator in $\mathbf{x}_t$.

We can rewrite this equation in the form of the continuity equation:

$$\frac{\partial p_t(\mathbf{x}_t)}{\partial t} = -\nabla \cdot \left(\tilde{\mathbf{v}}_t(\mathbf{x}_t)p_t(\mathbf{x}_t) - \frac{\sigma_t^2}{2}\frac{\nabla p_t(\mathbf{x}_t)}{p_t(\mathbf{x}_t)}p_t(\mathbf{x}_t)\right) \tag{16}$$

$$= -\nabla \cdot \left(\left(\tilde{\mathbf{v}}_t(\mathbf{x}_t) - \frac{\sigma_t^2}{2}\nabla\log p_t(\mathbf{x}_t)\right)p_t(\mathbf{x}_t)\right) \tag{17}$$

$$= -\nabla \cdot (\mathbf{w}_t p_t(\mathbf{x}_t)) \tag{18}$$

where the vector field $\mathbf{w}_t$ is defined as:

$$\mathbf{w}_t(\mathbf{x}_t) = \tilde{\mathbf{v}}_t(\mathbf{x}_t) - \frac{\sigma_t^2}{2}\nabla\log p_t(\mathbf{x}_t) = \sqrt{1 - \sigma_t^2}\mathbf{v}_t(\mathbf{x}_t) + \frac{2\gamma - 1}{2}\sigma_t^2\nabla\log p_t(\mathbf{x}_t) \tag{19}$$

As $t$ approaches 0 or 1, $\sigma_t \to 0$, causing $\tilde{\mathbf{v}}_t(\mathbf{x}_t)$ to converge to $\hat{\mathbf{v}}_t(\mathbf{x}_t)$. This ensures that the continuity equation is satisfied with the probability path $p_t(\mathbf{x}_t)$ for the original equation.

### A.1.2 TRAINING OBJECTIVE DERIVATIONS

As described in Theorem 2 of (Lipman et al., 2022), the FM loss is defined as:

$$\mathcal{L}_{\text{FM}}(\theta) = \mathbb{E}_{t,\mathbf{x}_1,p_t(\mathbf{x}_t|\mathbf{x}_1)}\big\|\mathbf{v}_t(\mathbf{x}_t) - \hat{\mathbf{v}}_t(\mathbf{x}_t)\big\|^2 \tag{20}$$

The CFM loss, as defined in equation 5, is equivalent to the FM loss up to a constant value, implying that $\mathcal{L}_{\text{FM}}(\theta) = \mathcal{L}_{\text{CFM}}(\theta)$. Based on this equivalence, we can express the training objective for approximating the vector field in equation 7, $\hat{f}(\mathbf{x}_t, t \mid \mathbf{x}_1, \mathbf{x}_0; \theta)$, as:

$$\mathcal{L}_{\text{IDFF}}^1(\theta) = \mathbb{E}_{t,\mathbf{x}_1,p_t(\mathbf{x}_t|\mathbf{x}_1)}\big\|\hat{\mathbf{v}}_t(\mathbf{x}_t \mid \mathbf{x}_1; \theta) - \mathbf{v}_t(\mathbf{x}_t|\mathbf{x}_1)\big\|^2 \tag{21}$$

To recover unbiased samples, we replace $\hat{u}(.)$ with equation 10 and $\hat{\mathbf{v}}_t(\mathbf{x}_t|\mathbf{x}_1, \mathbf{x}_0)$ with equation 9, resulting in:

$$\mathcal{L}_{\text{IDFF}}^1(\theta) = \mathbb{E}_{t,\mathbf{x}_1,p_t(\mathbf{x}_t|\mathbf{x}_1)}\left[\left\|\frac{\hat{\mathbf{x}}_1(\mathbf{x}_t, t; \theta) - \mathbf{x}_t}{1 - t} - \frac{\mathbf{x}_1 - \mathbf{x}_t}{1 - t}\right\|^2\right] \tag{22}$$

$$= \mathbb{E}_{t,\mathbf{x}_1,p_t(\mathbf{x}_t|\mathbf{x}_1)}\big[\beta(t)^2\big\|\hat{\mathbf{x}}_1(\mathbf{x}_t, t; \theta) - \mathbf{x}_1\big\|^2\big] \tag{23}$$

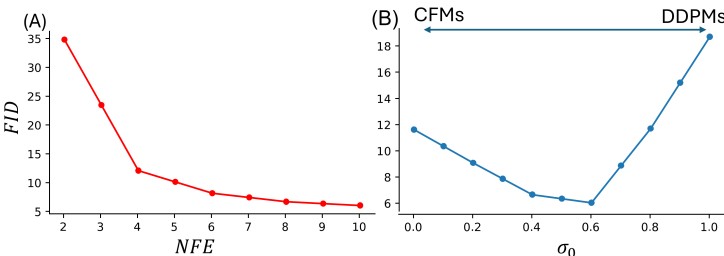

Figure 4: IDFF performance against A) the number of function evaluations (NFEs) and B) the hyperparameter $\sigma_0$ (the flow and score model mixing term) with fixed NFE=10 for the CIFAR-10 dataset. Similar analysis for CelebA-64 is shown in Figure5

where $\beta(t) = \frac{1}{1-t}$. Therefore, we have:

$$\mathcal{L}_{\text{IDFF}}^1(\theta) = \mathbb{E}_{t,\mathbf{x}_1,p_t(\mathbf{x}_t|\mathbf{x}_1)}\left[\beta(t)^2\left\|\hat{\mathbf{x}}_1(\mathbf{x}_t,t;\theta) - \mathbf{x}_1\right\|^2\right] \tag{24}$$

To approximate $\nabla_\mathbf{x} \log p_t(\mathbf{x}_t|\mathbf{x}_1)$, we can employ a time-dependent score-based model, $\epsilon(\mathbf{x}_t,t;\theta)$, using a continuous loss function with a weighting schedule $\lambda(t)$. Since $\nabla_\mathbf{x} \log p_t(\mathbf{x}_t|\mathbf{x}_1)$ approaches infinity as $t$ tends to 0 or 1, it is necessary to standardize the loss to maintain consistency over time. We set $\lambda(t)$ such that the target has zero mean and unit variance, predicting the noise added in sampling $\mathbf{x}_t$ before multiplying by $\sigma_t = \sigma_0\sqrt{t(1-t)}$. This leads to $\lambda(t) = \sigma_t = \sigma_0\sqrt{t(1-t)}$, ensuring that the regression target for $\hat{\epsilon}$ is distributed as $\mathcal{N}(\mathbf{0},\mathbf{I})$. Notably, this loss function is independent of the $\gamma$ value, allowing us to use the same model with different $\gamma$ values during the sampling process without altering the minima. The loss is defined as:

$$\mathcal{L}_{\text{IDFF}}^2(\theta) = \mathbb{E}_{t,\mathbf{x}_1,p_t(\mathbf{x}_t|\mathbf{x}_1)}\left[\lambda(t)^2\left\|\hat{\epsilon}(\mathbf{x}_t,t;\theta) - \nabla_\mathbf{x}\log p_t(\mathbf{x}_t|\mathbf{x}_1)\right\|^2\right] \tag{25}$$

where $\lambda(\cdot)$ is a set of positive weights. With a proper choice of $\lambda(t)$, this loss is equivalent to the original DDPM loss up to a constant value, according to Theorem 1 of (Song et al., 2020b). Combining the two components, the IDFF loss function is defined as:

$$\mathcal{L}_{\text{IDFF}}(\theta) = \mathcal{L}_{\text{IDFF}}^1(\theta) + \mathcal{L}_{\text{IDFF}}^2(\theta) \tag{26}$$

Which leads to the final objective function:

$$\mathcal{L}_{\text{IDFF}}(\theta) = \mathbb{E}_{t,\mathbf{x}_1,p_t(\mathbf{x}_t|\mathbf{x}_1)}\left[\beta(t)^2\|\hat{\mathbf{x}}_1(\mathbf{x}_t,t;\theta) - \mathbf{x}_1\|^2 + \lambda(t)^2\left\|\epsilon(\mathbf{x}_t,t;\theta) - \nabla_\mathbf{x}\log p_t(\mathbf{x}_t|\mathbf{x}_1)\right\|^2\right] \tag{27}$$

## A.2 BALANCING THE IDFF FLOW

IDFF consists of a scoring model and an implicit flow component coupled with a parameter $\sigma_0$ to generate flow $\mathbf{w}(.)$. We examine the effect of $\sigma_0$ on sample quality for the CIFAR-10 and CelebA datasets, with the NFE fixed at 10. As shown in Figure 4 (B), a value of $\sigma_0 = 0.6$ achieves the best balance between the diffusion and flow components regarding sample quality for both datasets. This plot also demonstrates that sample quality is susceptible to $\sigma_0$, particularly for diffusion-based models like DDPMs, compared to CFMs, especially at small NFEs, which is already noticed in some research(Jolicoeur-Martineau et al., 2020; Jain et al., 2020).

## A.3 ON CONVERGENCE SPEED OF IDFF

IDFF introduces a momentum term that slightly increases training complexity compared to CFMs. However, this added complexity is offset by a significant reduction in the number of sampling steps required during inference. Leveraging the strengths of its underlying CFM models, IDFF achieves high-quality sample generation (FID < 10) within just 500K training iterations, as demonstrated in Figure 6, while maintaining competitive performance with CFMs.

Figure 6 shows the FID curves during IDFF training in the CIFAR-10 dataset with a batch size of 128, emphasizing its superior efficiency compared to conventional diffusion models. This efficiency

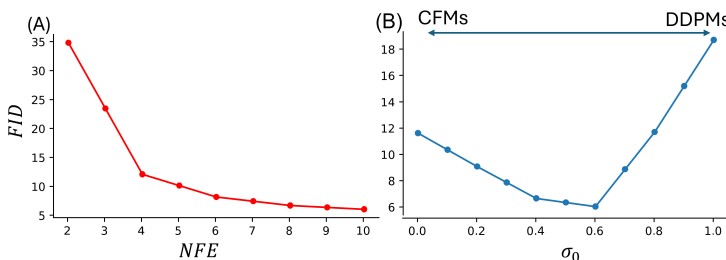

Figure 5: IDFF performance against A) the number of function evaluations (NFEs) and B) the hyperparameter $\sigma_0$ (the flow and score model mixing term) with fixed NFE=10, for CelebA-64 dataset.

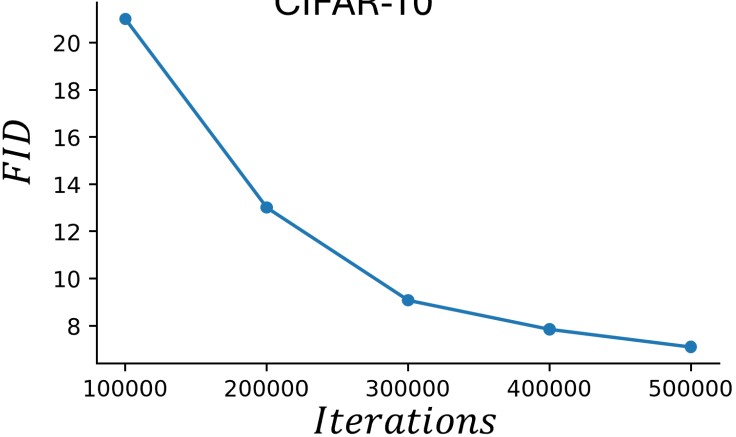

Figure 6: IDFF performance against the number of training iterations with fixed NFE=10 for the CIFAR-10 dataset.

marks a substantial improvement over the millions of iterations typically required by diffusion-based models to generate high-quality images, which entail considerable computational costs. In contrast, as depicted in Figure 6, IDFF retains the training efficiency of CFMs, achieving faster convergence and significant computational savings.

## A.4 IDFF TRAINING AND SAMPLING ALGORITHMS FOR TIME SERIES

---

**Algorithm 3** IDFF sampling algorithm for time-series data.

---

**Input:** $\hat{\mathbf{x}}_1(.;\theta)$ and $\hat{\boldsymbol{\epsilon}}(.;\theta)$, maximum sequence length $K$, bandwidth $\sigma_0$, time step size $\Delta t$, and $\gamma$.
**for** $k$ in $\{1,...,K\}$ **do**
    **if** $k == 1$ **then**
        $\mathbf{x}_0^0 \sim \mathcal{N}(0, \mathbf{I})$
    **else**
        $\mathbf{x}_0^k \sim \mathcal{N}(\mathbf{x}_1^{k-1}, \sigma_0 \mathbf{I})$
    **end if**
    **for** $t$ in $[0, 1/\Delta t)$ **do**
        $\sigma_t \leftarrow \sigma_0 \sqrt{t(1-t)}$
        $\mathbf{w}_t^k \leftarrow \sqrt{(1-\sigma_t^2)}\frac{\hat{\mathbf{x}}_1(\mathbf{x}_t,t,k;\theta)-\mathbf{x}_t^k}{1-t} + \frac{2\gamma-1}{2}\sigma_t\hat{\epsilon}(\mathbf{x}_t, t, k; \theta)$
        $\mathbf{x}_{t+\Delta t}^k \sim \mathcal{N}(\mathbf{x}_t^k + \mathbf{w}_t^k \Delta t, \sigma_t^2 \Delta t \mathbf{I})$
    **end for**
**end for**
**return** Samples $\{\mathbf{x}_1^k\}_{k=1}^{k=K}$

---

---

**Algorithm 4** IDFF training algorithm for time-series data.

---

**Input:** dataset distribution $p_1(\mathbf{x}_1)$, initial distribution $p_0(\mathbf{x}_0)$, maximum sequence length $K$, bandwidth $\sigma_0$, a set of positive weights $\lambda(.)$ and $\beta(.)$, initialized networks $\hat{\mathbf{x}}_1(.;\theta)$ and $\hat{\boldsymbol{\epsilon}}(.;\theta)$,
**while** Training **do**
    **if** $K == 1$ **then**
        $k = 1$
        $\mathbf{x}_1 \sim p_1(\mathbf{x}_1), \mathbf{x}_0 \sim p_0(\mathbf{x}_0)$
    **else**
        $k \sim \{1,...,K\}$ with equal probability
        $\mathbf{x}_1 \sim p_1(\mathbf{x}_1^k), \mathbf{x}_0 \sim p_1(\mathbf{x}_1^{k-1})$
    **end if**
    $\pi \leftarrow \text{OT}(\mathbf{x}_1, \mathbf{x}_0)$
    $(\mathbf{x}_0, \mathbf{x}_1) \sim \pi$
    $t \sim \mathcal{U}(0,1)$
    $\boldsymbol{\mu}_t \leftarrow t\mathbf{x}_1 + (1-t)\mathbf{x}_0$
    $\sigma_t \leftarrow \sigma_0 \sqrt{t(1-t)}$
    $\mathbf{x}_t \sim \mathcal{N}(\boldsymbol{\mu}_t, \sigma_t^2 \mathbf{I})$
    $\mathcal{L}_{\text{IDFF}}(\theta) \leftarrow \beta(t)^2 \|\hat{\mathbf{x}}_1(\mathbf{x}_t, t, k; \theta) - \mathbf{x}_1\|^2 + \lambda(t)^2 \|\boldsymbol{\epsilon}(\mathbf{x}_t, t; \theta) - \nabla_x \log p_t(\mathbf{x}_t|\mathbf{x}_1)\|^2$
    $\theta \leftarrow \text{Update}(\theta, \nabla_\theta L(\theta))$
**end while**
**return** $\{\hat{\mathbf{x}}_1(.;\theta), \hat{\epsilon}(.;\theta)\}$

---

## A.5 IMPLEMENTATION DETAILS

**Network configuration:** We adopt UNet (ADM) for our image generation and SST forecasting experiments. Table 5 shows detailed configurations of the ADM network on different datasets.

**Training hyper-params.** In Table 6, we provide training hyperparameters for unconditional image generation on ADM and SST forecasting datasets.

Table 5: ADM network configuration for different datasets.

|  | CIFAR-10 | CelebA64 | CelebA 256 | Church & Bed | SST |
|---|---|---|---|---|---|
| # of ResNet blocks | 2 | 2 | 2 | 2 | 2 |
| Base channels | 128 | 128 | 128 | 256 | 128 |
| Channel multiplier | 1,2,2,4 | 1,2,2,4,4 | 1,1,2,2,4,4 | 1,1,2,2,4,4 | 1,2,4 |
| Attention resolutions | 16 | 16 | 16 | 16 | 16 |
| Label dimensions | 1 | 1 | 1 | 1 | 10 |
| Params (M) | 65.6 | 102.14 | 453.45 | 108.41 | 55.39 |

Table 6: Hyper-parameters of ADM network.

|  | ImageNet-64 | CIFAR-10 | CelebA64 | CelebA 256 | Church & Bed | SST |
|---|---|---|---|---|---|---|
| lr | 1e-4 | 1e-4 | 1e-5 | 1e-5 | 1e-5 | 1e-4 |
| Batch size | 64 | 128 | 128 | 16 | 16 | 8 |
| # of iterations | 3M | 700K | 1.2M | 1.5M | 1.5M | 700K |
| # of GPUs | 4 | 1 | 1 | 1 | 1 | 1 |

## A.6 ADDITIONAL RESULTS FOR IMAGE GENERATION EXPERIMENT

We also assesd IDFF performance in generating images against fast diffusion process models with MFE=5. As Table7 IDFF achieves a significantly better FID score (10.97) compared to UniPC (23.52) and DPM-Solver-v3 (12.21), while also boasting the fastest wall-clock time (0.34 seconds) among all solvers. This highlights IDFF's ability to generate high-quality samples with minimal computational overhead, making it ideal for real-time applications. Even at NFE=10, where UniPC slightly edges out in FID (2.85) alongside DPM-Solver-v3 (2.91), IDFF remains competitive with a reasonable FID (5.87) and the fastest wall-clock time (0.52 seconds), demonstrating its efficiency and scalability. These results suggest that IDFF hits a balance between sample quality and computational speed that lends itself to speed.

Table 7: Comparison of FID↓ performance and sampling times (Wall-clock) between IDFF and fast diffusion sampling methods for NFE=5 and NFE=10, evaluated on 50k samples.

| Method | FID (NFE=5) | Wall-clock (sec, NFE=5) | FID (NFE=10) | Wall-clock (sec, NFE=10) |
|---|---|---|---|---|
| UniPC (Zhao et al., 2024) | 23.52 | 0.62 | **2.85** | 1.05 |
| DPM-Solver-v3 (Zheng et al., 2023) | 12.21 | 0.49 | 2.91 | 0.92 |
| IDFF | **10.97** | **0.34** | 5.87 | **0.52** |

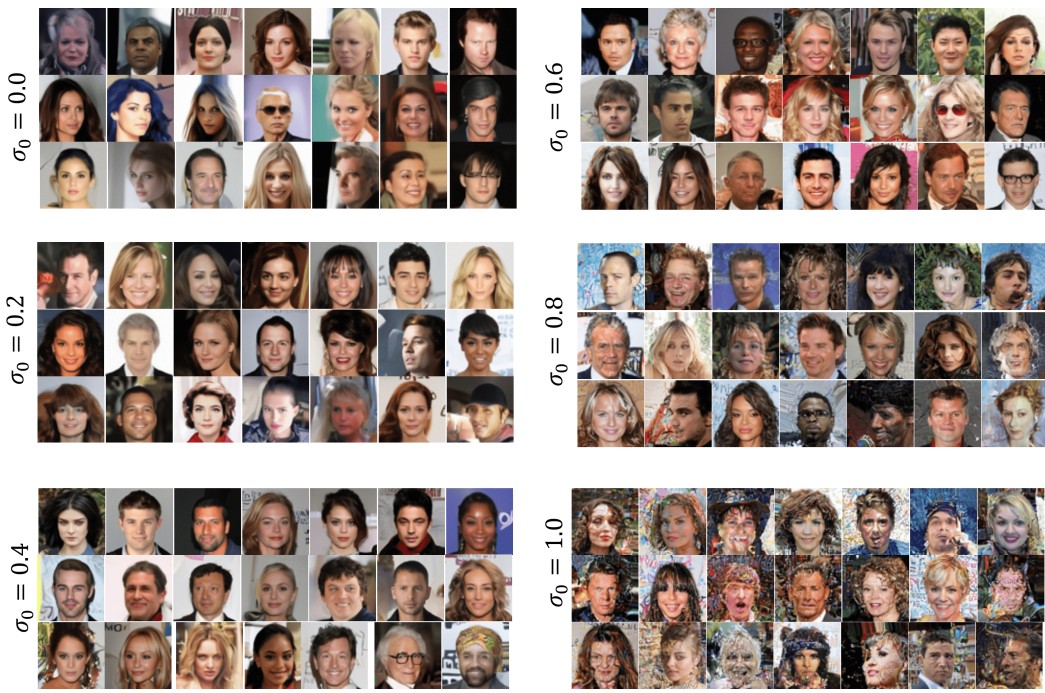

Figure 7: Generated samples for CelebA64 ($64 \times 64$) dataset with different $\sigma_0$s and $NFE = 10$.

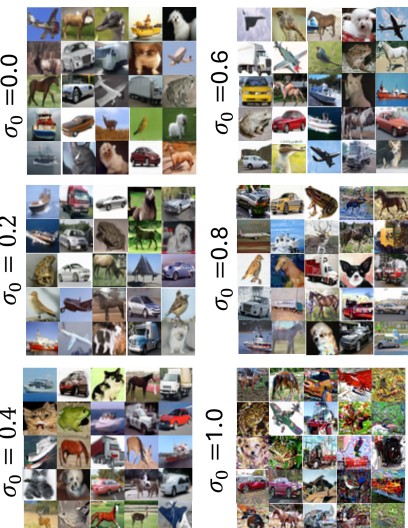

Figure 8: Generated samples for CIFAR-10 ($32 \times 32$) dataset with different $\sigma_0$s and $NFE = 10$.

## A.7 3D-ATTRACTORS

In this experiment, we assess IDFF's performance in generating trajectories of chaotic systems from scratch. We generate trajectories with $K = 2000$ samples in each trajectory from 3D attractors, specifically the Lorenz and Rössler attractors, which are chaotic systems with nonlinear dynamics. The parameters for the Lorenz and Rössler models are set to $\sigma = 10, \rho = 28, \beta = 8/3$ and $a = .2, b = .2, c = 5.7$, respectively, to produce complex trajectories in 3D space. We then train the IDFF model based on these trajectories. To model each attractor, we use an MLP with two hidden layers of 128 dimensions and two separate heads for $\hat{\mathbf{x}}_1(., t, k; \theta)$ and $\epsilon(\mathbf{x}_t, t, k; \theta)$. Additionally, we incorporate two separate embedding layers for embedding $t$ and $k$, which are directly concatenated with the first hidden layer of the MLP. The optimized IDFF successfully generates samples of these

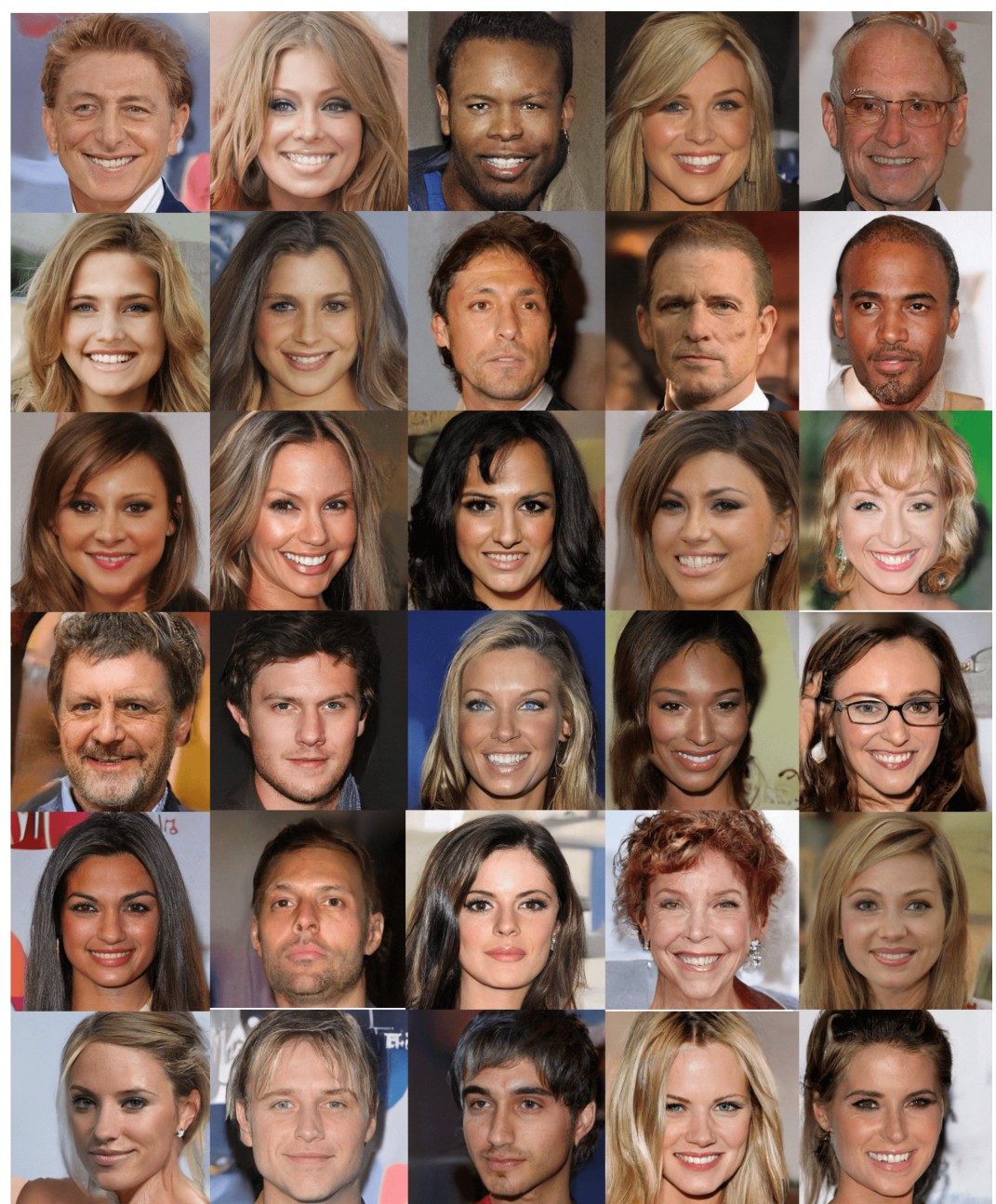

Figure 9: Generated samples for CelebA-HQ ($256 \times 256$) dataset with $\sigma_0 = 0.2$ and $NFE = 10$.

trajectories from scratch. The generated trajectories are shown in Figure 13. The quality of the results demonstrates that IDFF can successfully simulate the behaviors of highly nonlinear and nonstationary systems such as attractors.

## A.8 SST FORECASTING VISUALIZATION

## A.9 2D-SIMULATED STATIC DATA AND TIME-SERIES

Additional results for 2D simulations for both static and time-series generation are shown in Figure16.

1134
1135
1136
1137
1138
1139
1140
1141
1142
1143
1144
1145
1146
1147
1148
1149
1150
1151
1152
1153
1154
1155
1156
1157
1158
1159
1160
1161
1162
1163
1164
1165
1166
1167
1168
1169
1170
1171
1172
1173
1174
1175
1176
1177
1178
1179
1180
1181
1182
1183
1184
1185
1186
1187

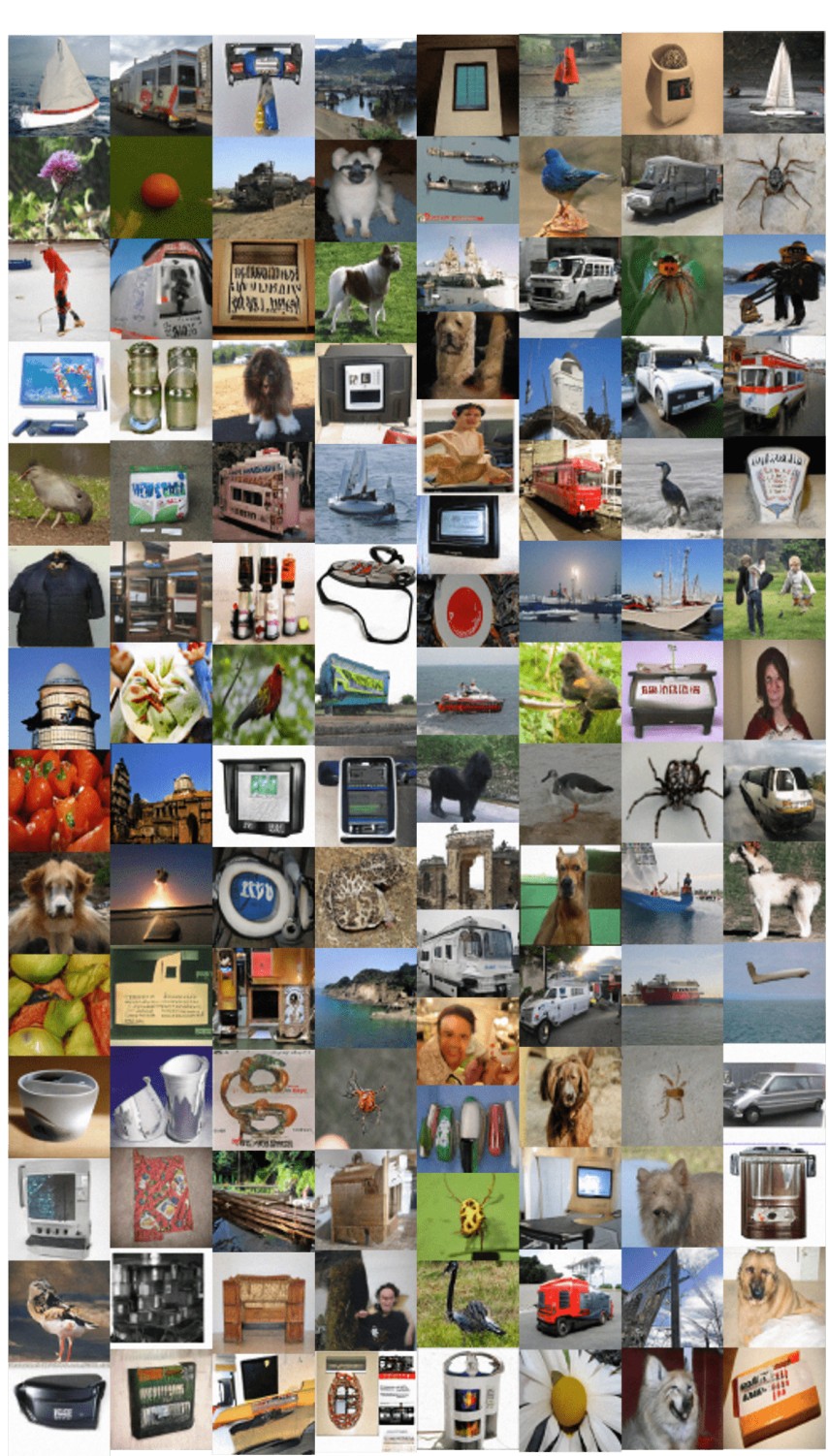

Figure 10: Generated samples for ImageNet64 dataset with $\sigma_0 = 0.2$ and $NFE = 10$.

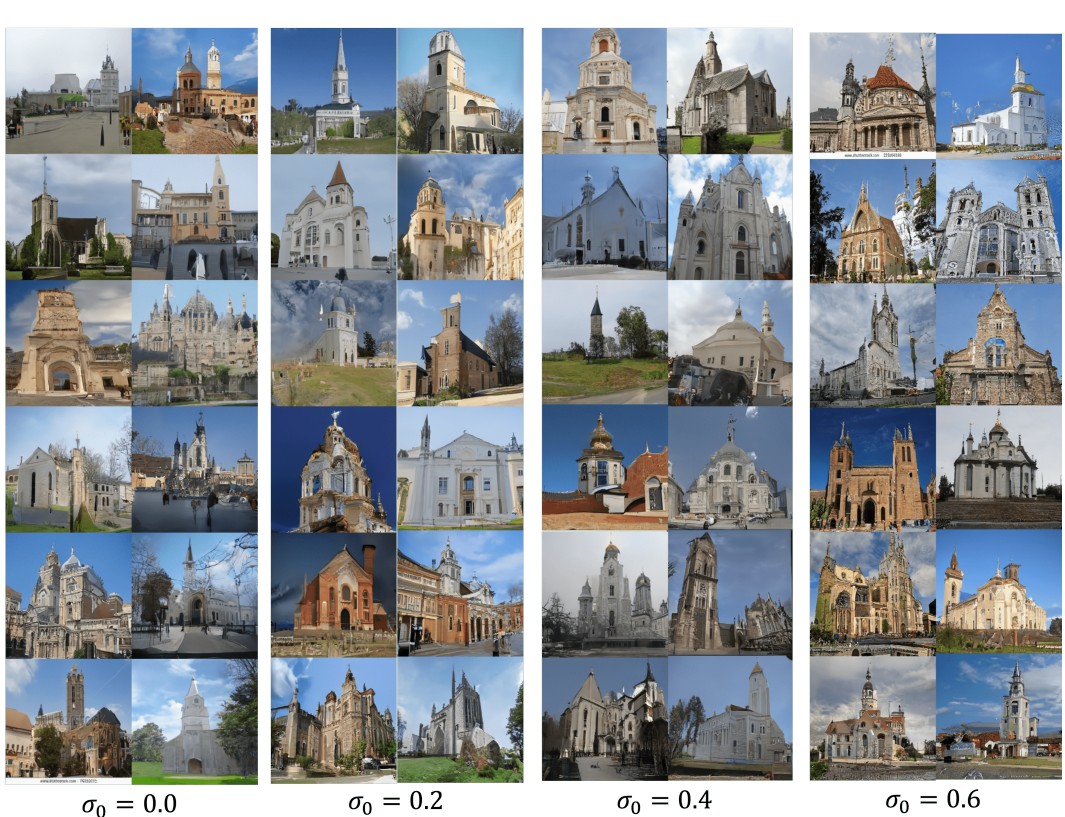

$\sigma_0 = 0.0$ $\qquad\qquad$ $\sigma_0 = 0.2$ $\qquad\qquad$ $\sigma_0 = 0.4$ $\qquad\qquad$ $\sigma_0 = 0.6$

Figure 11: Generated samples for LSUN-church ($256 \times 256$) dataset with different $\sigma_0$s and $NFE = 10$.

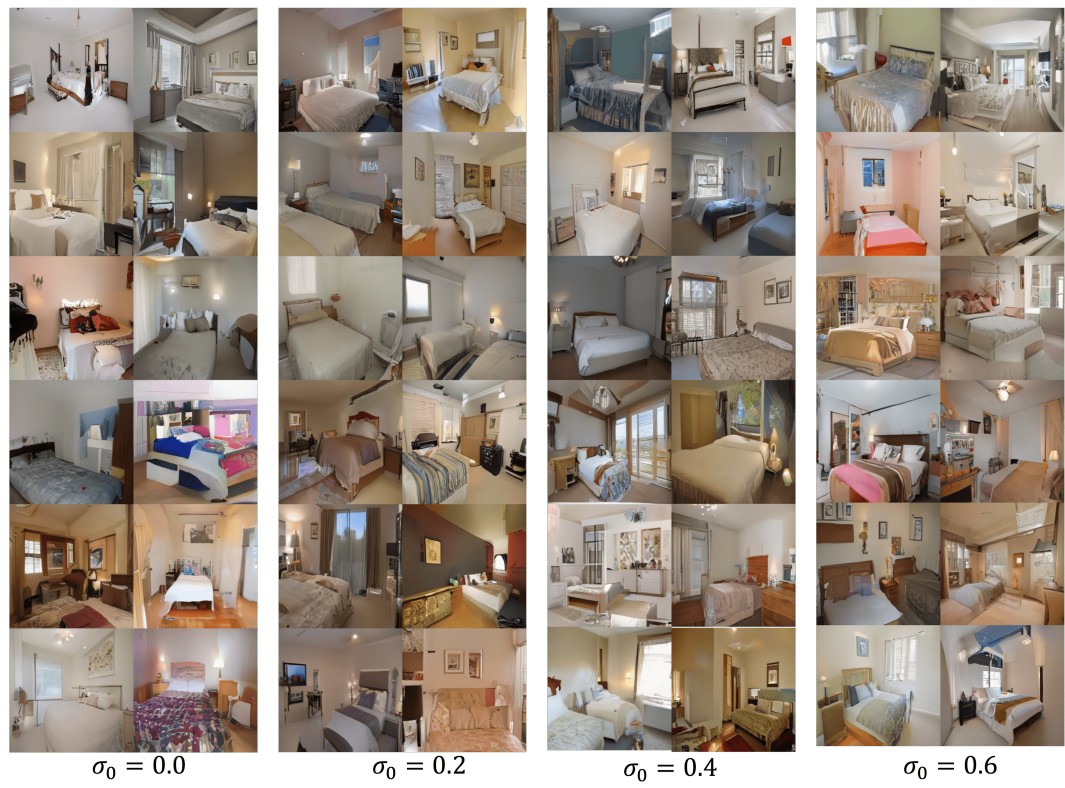

$$\sigma_0 = 0.0 \qquad \sigma_0 = 0.2 \qquad \sigma_0 = 0.4 \qquad \sigma_0 = 0.6$$

Figure 12: Generated samples for LSUN-bed ($256 \times 256$) dataset with different $\sigma_0$s and $NFE = 10$.

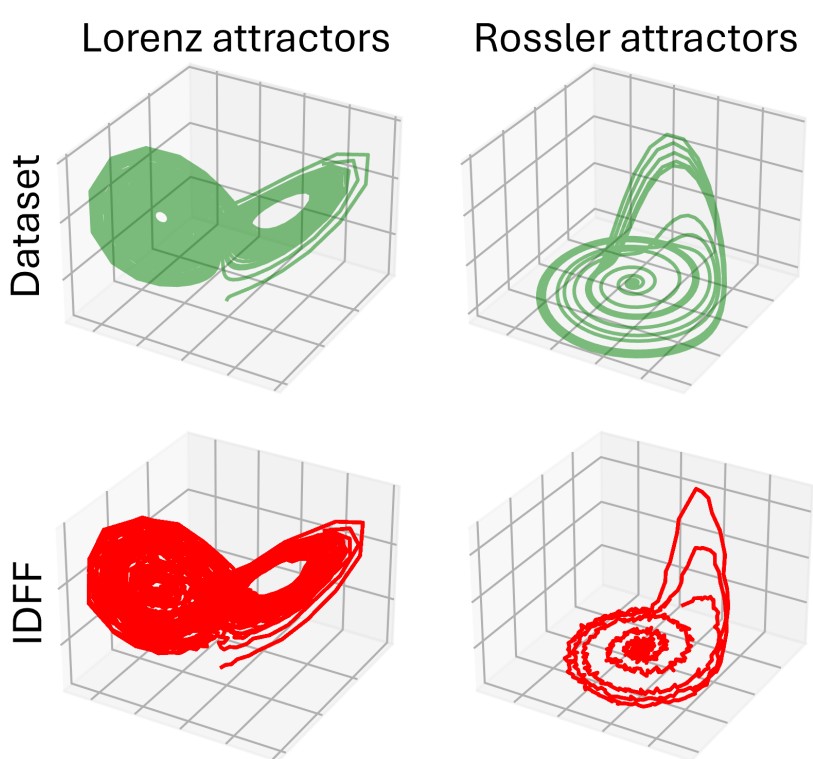

Figure 13: Time-series simulation. IDFF trajectory generation for the chaotic systems.

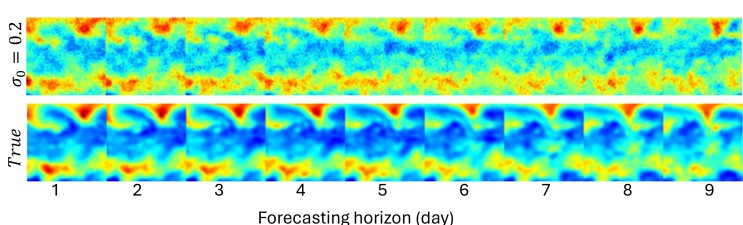

Figure 14: SST forecasting result conditioned on day 1st for 9 days with $\sigma_0 = 0.2$ and fixed $NFE = 5$. Same results for different $\sigma_0$s is shown in Figure15

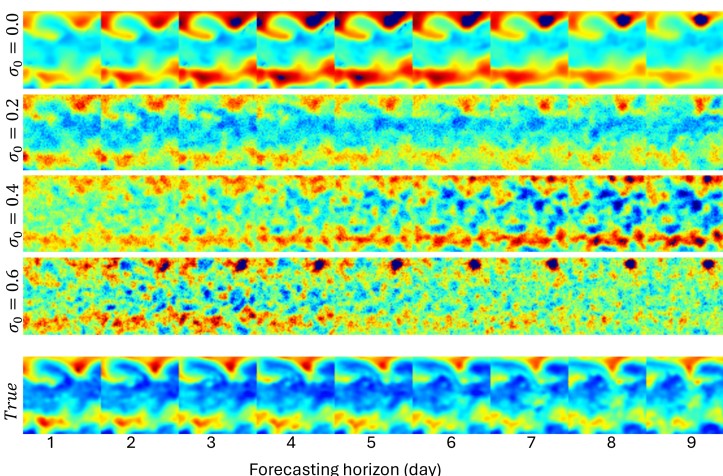

Figure 15: SST forecasting result conditioned on day 1st for 9 days for different values of $\sigma_0$ and fixed $NFE = 5$.

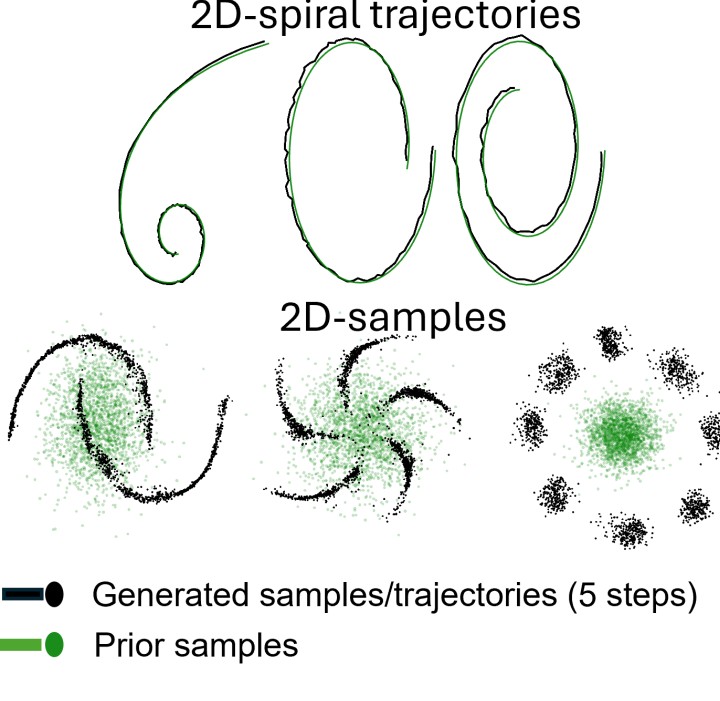

Figure 16: 2D synthetic simulation.

