# OpenReview forum: "Implicit Dynamical Flow Fusion (IDFF) for Generative Modeling"
_ICLR.cc/2025/Conference — Submitted to ICLR 2025_

### Official Review · Reviewer_vf9H · 2024-10-21

**Soundness:** 2
**Presentation:** 2
**Contribution:** 3
**Rating:** 6
**Confidence:** 4

**Summary:**

The paper introduces Implicit Dynamical Flow Fusion (IDFF) for generative modeling. IDFF learns a vector field with an additional momentum term, allowing for the use of larger step sizes during sampling without compromising the quality of the generated data, thereby accelerating the generation process. This method has demonstrated effectiveness across different domains, including image generation and time-series datasets modeling.

**Strengths:**

The paper proposes a novel approach for training Conditional Flow Matching (CFM), which adds an additional correction term to the vector field, allowing for larger step sizes during the sampling process, thereby accelerating sampling. The paper also conducts extensive experiments to validate the effectiveness of the method, including tasks in image generation and time-series datasets modeling.

**Weaknesses:**

1. Some parts of the paper need improvement in their descriptions. For instance, in Section 3.1, during the transition from $\tilde{\mathbf{v}}_t\left(\mathbf{x}_t\right)$ to $\mathbf{w}_t\left(\mathbf{x}_t\right)$, a stochastic differential equation (SDE) is introduced (in Appendix A.1.1), but this is not explicitly mentioned in the main text, nor is its purpose clearly explained. Perhaps the introduction of the SDE here is intended to eliminate the effect of introducing the momentum term on the probability density path $p_t(\mathbf{x}_t)$?
2. Some experiments in the paper are insufficient. For example, in the image generation task, there is no comparison with some classic Conditional Flow Matching (CFM) methods, such as rectified flow[1].

References:
[1] Xingchao Liu, Chengyue Gong, and Qiang Liu. Flow straight and fast: Learning to generate and transfer data with rectified flow. arXiv preprint arXiv:2209.03003, 2022.

**Questions:**

1. Could you provide a clearer explanation of the transition from $\tilde{\mathbf{v}}_t\left(\mathbf{x}_t\right)$ to $\mathbf{w}_t\left(\mathbf{x}_t\right)$ in Section 3.1 of the paper?

---

> ### Author Response · Authors · 2024-11-21
>
> Thank you for your detailed review and valuable feedback. We appreciate your recognition of IDFF’s novelty and effectiveness and your suggestions for enhancing clarity and comprehensiveness.
>
> ***Some parts of the paper need improvement in their descriptions. For instance...***
>
>
> The introduction of the SDE (detailed in Appendix A.1.1) primarily supports the proof of continuity. The solutions $\boldsymbol{x}_t$ of this SDE are governed by the Fokker-Planck equation, which is instrumental in the proof process. Importantly, as shown in [2], the chosen SDE shares the same marginal distributions as the probability ODE paths, ensuring consistency in the marginal distribution.
> Your observation is correct—the SDE plays a key role in demonstrating that the introduction of the momentum term does not ultimately affect the marginal probability density path $p_t(\boldsymbol{x}_t)$.
>
> **Could you provide a clearer explanation of the transition..**
>
> While $\tilde{\boldsymbol{v}}_t$ provides flexibility by incorporating momentum, it does not inherently satisfy the probability flow constraint, which requires alignment with the continuity equation. To ensure this constraint is met, the IDFF framework refines $\tilde{\boldsymbol{v}}_t$ into a new vector field $\boldsymbol{w}_t$. This refinement allows IDFF to retain the flexibility provided by leveraging momentum while ensuring that the theoretical constraints required for flow matching (continuity constraint) remain valid and consistent throughout. We will enhance Section 3 to provide clearer explanations and context for the transition from $\tilde{\boldsymbol{v}}_t$ to $\boldsymbol{w}_t$.
>
>
>
> ```[2] Song, Y., Sohl-Dickstein, J., Kingma, D. P., Kumar, A., Ermon, S., & Poole, B. (2020). Score-based generative modeling through stochastic differential equations. arXiv preprint arXiv:2011.13456.```
>
> ***Some experiments in the paper are insufficient. For example, in the image generation task...***
>
> We appreciate your suggestion to benchmark IDFF against Rectified Flow and will include a comparison and the proper citation in our analysis of the image generation task. Moreover, we recognize that [1] shares a robust conceptual connection with Independent Coupling Flow Matching (I-CFM). Specifically, when the conditional probability path p_t​ is a Dirac delta function (i.e., σ=0), I-CFM becomes equivalent to Rectified Flow. In this case, the reported FID score is 13.68 for Rectified Flow with NFE=10 [1]. We have also computed the FID score for simple FM, equal to  14.36, with NFE=10. These results will be added to the revised version of the paper to ensure a more thorough comparison.
>
> These revisions will enhance the paper’s clarity and strengthen the experimental results, addressing your concerns comprehensively.
>
> Thank you again for your constructive feedback; please let us know if we have appropriately answered your questions and if you have any follow-ups on the same.

---

> > ### Author Response · Authors · 2024-11-27
> >
> > We believe that we have addressed all the concerns raised in your review. We have clarified the key areas you pointed out, improved the paper’s structure and descriptions, and included additional experimental results in the revised manuscript and in this response.
> >
> > In response to your valuable feedback, we have made the following updates:
> >
> > ### **Clarification on the Introduction of the SDE and Its Role**
> > We have clarified the introduction of the Stochastic Differential Equation (SDE) to emphasize its critical role in supporting the proof of continuity. Specifically, we detailed how the solutions of this SDE share the same marginal distributions as the probability ODE paths, ensuring consistency in the marginal distributions.
> >
> > ### **Enhanced Explanation of the Transition**
> > We have improved the explanation of how this transition ensures theoretical consistency while maintaining the flexibility to leverage momentum. This update aligns the explanation with the theoretical requirements of flow matching and continuity constraints.
> >
> > ### **Additional Experimental Results for the Image Generation Task**
> > In response to your suggestion, we benchmarked IDFF against Rectified Flow and Simple FM. We have also included Fréchet Inception Distance (FID) scores to provide a thorough comparison:
> >
> > - **Rectified Flow**: FID = 13.68 (NFE = 10)
> > - **Simple FM**: FID = 14.36 (NFE = 10)
> >
> > These results, now part of the revised manuscript, strengthen our experimental analysis and provide a more robust evaluation of IDFF.

---

> > > ### Comment · Reviewer_vf9H · 2024-11-28
> > >
> > > Thanks for the authors' responses. Some of my concerns have been addressed, but I still have a few questions.
> > >
> > > I understand that the ODE corresponding to $w_t$ is the Probability Flow ODE of the SDE corresponding to $\tilde{v}_t$, which means that their marginal probability densities $p_t$ are consistent. However, how does the ODE corresponding to $w_t$ maintain the same marginal probability density $p_t$ as the ODE corresponding to $v_t$? The paper only uses the fact that $\tilde{v}_t$ converges to $v_t$ as $t$ approaches 0 and 1 (Lines 282-284) to address this issue, but I believe this argument is insufficient, because the two time-varying vector fields being identical only at the initial and final points clearly does not guarantee the consistency of the marginal probability densities. If there is an error in the above question, I would appreciate it if the authors could point it out; otherwise, I would like to request a more rigorous proof to ensure that the ODE corresponding to $\tilde{v}_t$ and the ODE corresponding to $v_t$ have the same marginal probability density $p_t$.
> > >
> > > My understanding is that the momentum term $\xi_t$ in the paper will guide the particle to move towards regions of higher probability density. However, does this approach potentially lead to mode collapse in the final generated results? I would appreciate it if the authors could compute the precision and recall values of IDFF in the image generation task as a reference.[1]
> > >
> > > [1] Kynkäänniemi, T., Karras, T., Laine, S., Lehtinen, J., & Aila, T. (2019). Improved precision and recall metric for assessing generative models. Advances in neural information processing systems, 32.

---

> > > > ### Author Response · Authors · 2024-11-28
> > > >
> > > > **I understand that the ODE corresponding to wt is the Probability Flow ODE of the SDE corresponding to v~t, which means that their marginal probability densities pt are consistent...**
> > > >
> > > > This may be a misunderstanding related to which vector is being matched by our proposal.
> > > >
> > > > To clarify, we care about maintaining the same marginal probability $p_t(x)$ because it ensures that we are sampling from the data density. As you mentioned, the design of $\tilde{\mathbf{v}}_t$ converges to $\mathbf{v}_t$ as $t$ approaches 0 and 1. You are right that this alone is insufficient to guarantee the consistency of the marginal probability densities. However, IDFF optimizes components of $\mathbf{w}_t$, which by design (c.f. Equation (7)) means that it matches $\mathbf{v}_t$ at 0 and 1. Furthermore, in Lemma 1, we assume $\mathbf{v}_t$ is the vector field that generates the probability path $p_t(x)$. With this assumption, we show that $\mathbf{w}_t$ also satisfies the continuity equation for $p_t(x)$ (Equation 18 in the appendix). The combination of these facts ensures that the marginals under IDFF are consistent with $p_t(x)$.
> > > >
> > > > To expand on the above explanation further, for the ODE corresponding to $\mathbf{w}_t$ to maintain the same $p_t$, it must ensure that the flow of probability through space behaves consistently with $p_t$. This is guaranteed by the \emph{continuity equation}, which ensures that the flow of probability neither creates nor destroys mass as it moves.
> > > > The new flow $\mathbf{w}_t$ combines two components:
> > > >
> > > > The original flow $\mathbf{v}_t$.
> > > > A "momentum term" $\boldsymbol{\xi}_t$ is designed to guide the flow based on the log-probability gradient of $p_t$.
> > > > Mathematically, we define:
> > > > $$
> > > > \mathbf{w}_t = \sqrt{1 - \sigma_t^2} \mathbf{v}_t + \frac{2\gamma - 1}{2} \sigma_t^2 \nabla \log p_t(\mathbf{x}_t),
> > > > $$
> > > > where $\sigma_t$ is a time-dependent factor balancing these contributions. The term $\nabla \log p_t(\mathbf{x}_t)$, the \emph{momentum term} in IDFF, pushes the flow toward regions where $p_t$ is higher.
> > > > The continuity equation tells us how the probability density $p_t$ changes as the system evolves. For a flow $\mathbf{w}_t$, it is written as:
> > > > $$
> > > > \frac{\partial p_t}{\partial t} = -\nabla \cdot (\mathbf{w}_t p_t),
> > > > $$
> > > >
> > > > where $\nabla \cdot$ measures how much "outflow" or "inflow" the vector field $\mathbf{w}_t$ creates. Integrating the continuity equation over all space shows that:
> > > >
> > > > $$\frac{d}{dt} \int p_t(\mathbf{x}) d\mathbf{x} = 0. $$
> > > > Thus, the total probability $\int p_t(\mathbf{x}) , d\mathbf{x} = 1$ remains unchanged over time. This equation ensures that the total probability stays consistent over time because it represents the conservation of probability mass within the system. Think of $p_t$ as a fluid density and $\mathbf{w}_t p_t$ as the flow of the fluid. The continuity equation ensures that the fluid (probability) is neither created nor destroyed. It can only move from one region to another. Just as the total amount of fluid in a closed system remains constant, the total probability is conserved across the entire domain.
> > > > Appendix A demonstrates that $\mathbf{w}_t$ satisfies the continuity equation, ensuring that the marginal $p_t$ for $\mathbf{w}_t$ matches the original $p_t$. This guarantees that $p_t$ evolves correctly under $\mathbf{w}_t$. Transitioning to $\mathbf{w}_t$ introduces flexibility via the momentum term $\boldsymbol{\xi}_t$, enhancing the sampling process. Lemma 1 confirms that $\mathbf{w}_t$ preserves the marginal distribution $p_t$, ensuring accuracy while improving efficiency.
> > > >
> > > > **However, does this approach potentially lead to mode collapse in the final generated results?**
> > > >
> > > > We appreciate the reviewer’s observation that the momentum term $\boldsymbol{\xi}_t$ guides the flow toward regions of higher probability density, potentially raising concerns about mode collapse. One critical requirement is that $\mathbf{w}_t$ must reduce to $\mathbf{v}_t$ at the boundaries ($t = 0$ and $t = 1$). This requirement is met because the factor $\sigma_t \to 0$ at these points causes the contribution of the momentum term $\boldsymbol{\xi}_t$ to vanish. Consequently, $\mathbf{w}_t$ behaves identically to $\mathbf{v}_t$ at these boundaries, ensuring consistency with the original flow. Empirically, we have observed no evidence of mode collapse in the experiments conducted, as demonstrated by the diverse samples shown in Figures 7–12.

---

> ### Comment · Reviewer_vf9H · 2024-11-29
>
> Thanks for the author's response, but I believe it does not address my concerns.
>
> Regarding my first question, the key statement in the author's response is: “Furthermore, in Lemma 1, we assume $v_t$ is the vector field that generates the probability path $p_t$.” This directly assumes that $v_t$ and $\tilde{v}_t$ have the same marginal probability distribution, which is a strong assumption. The authors should provide further clarification to support this point. From the phrase “To expand on the above explanation further,” the author's response primarily elaborates on the properties of the continuity equation itself, namely, “the flow of probability neither creates nor destroys mass as it moves.” However, I do not believe this adequately addresses my concern about how $v_t$ and $\tilde{v}_t$ can produce the same marginal probability density $p_t$.
>
> For my second question, I suggest the authors calculate the precision and recall to quantitatively demonstrate that IDFF does not suffer from mode collapse. I do not believe the generated images in Figures 7–12 are sufficient to prove the absence of mode collapse in IDFF.
>
> Additionally, similar to Reviewer U2S3, I also have questions about how IDFF leverages HMC to reduce NFE. The statement in lines 215–217 of the paper, “These properties lead to faster convergence to the target distribution with fewer function evaluations,” lacks direct evidence. The properties of HMC, such as “reducing random walk behavior and overcoming local energy barriers,” are not clear to me in terms of how they directly reduce the number of steps required to solve the ODE numerically.
>
> This work is not convincing enough, both in terms of motivation and theoretical proof. I believe the authors need to present their motivation more clearly and provide a more rigorous proof for their theory.

---

> ### Author Response · Authors · 2024-12-01
>
> **However, I do not believe this adequately addresses my concern about how vt and v~t can produce the same marginal probability density pt.**
>
> Thank you for highlighting this concern. For $v_t(x)$ to produces the marginal density $p_t(x)$ it must satisfy:
>
> \begin{equation}
> v_t(x) = \mathbb{E}\_{q(z)} \left[\frac{v_t(x|z) p_t(x|z)}{p_t(x)}\right]. \tag{**}
> \end{equation}
>
> To show that $\tilde{v}_t(x)$ satisfies the same marginal probability we can write $\tilde{v}$ as a function of $v$, substitute it into the above, and obtain a condition on $\xi$. We can then show that $\xi=\gamma\log p_t(x)$ satisfies the necessary condition.
> **The full proof is provided in the following comment**. We will include the proof in Appendix A.1.1 of the manuscript to explicitly showcase that the transformed vector field $ \tilde{v}_t(x) $, defined by Equation (6), produces the same marginal probability density $ p_t(x) $ as the original vector field $ v_t(x) $.
>
> We hope this clarifies the concern about the theoretical rationale behind our work. We will work to incorporate this argument better in the main paper as well.
>
> **For my second question, I suggest the authors calculate the precision and recall to quantitatively demonstrate that IDFF does not suffer from mode collapse.**
>
> We appreciate the reviewer's suggestion to calculate precision and recall as quantitative measures to assess whether IDFF suffers from mode collapse. In response, we evaluated IDFF on CIFAR-10 with NFE=10, resulting in a precision of 0.76 and a recall of 0.54. These results indicate that IDFF effectively avoids mode collapse, as a high recall demonstrates the model's ability to generate diverse samples that cover the true data distribution.
>
> For comparison, recall values from other diffusion-based models under similar evaluation settings are as follows (adapted from [4]):
>
> DDIM with NFE=50: Recall = 0.53
>
> DDPM with NFE=1000: Recall = 0.57
>
> Notably, IDFF achieves a higher recall than DDIM despite requiring significantly fewer function evaluations (NFE=10 compared to NFE=50). This highlights IDFF's robustness in maintaining diversity across generated samples.
>
> [4] Zheng, Huangjie, et al. "Truncated diffusion probabilistic models and diffusion-based adversarial auto-encoders." arXiv preprint arXiv:2202.09671 (2022).

---

> > ### Author Response · Authors · 2024-12-01
> >
> > **Full Proof and Appendix A.1.1**
> >
> > We demonstrate that the transformed vector field $ \tilde{v}_t(x) $, defined by Equation (6), produces the same marginal probability density $ p_t(x) $ as the original vector field $ v_t(x) $. We proceed by explicitly starting from established assumptions in Conditional Flow Matching (CFM) and derive the necessary conditions step-by-step.
> >
> > Assumptions:
> >
> > 1-The vector field $ v_t(x) $ evolves the marginal probability density $ p_t(x) $ over time.
> >
> > 2-The  $ v_t(x|z) $ evolves the $ p_t(x|z) $, where $ z $ is an auxiliary random variable distributed according to a prior $ q(z) $ usually distributed uniformly.
> >
> > 3-The $ v_t(x) $ satisfies the following relationship with respect to the $ v_t(x|z) $ via:
> >
> > $$v_t(x) = \mathbb{E}\_{q(z)} \left[\frac{v_t(x|z) p_t(x|z)}{p_t(x)}\right]. \tag{**}$$
> >
> >
> > Goal:
> > We aim to show that the transformed vector field $ \tilde{v}_t(x) $, defined as:
> > \begin{equation*}
> > \tilde{v}_t(x) = \sqrt{1 - \sigma_t^2} v_t(x) + \sigma_t^2 \xi_t,
> > \end{equation*}
> > produces the same marginal probability density $ p_t(x) $ under specific conditions on $ \xi_t $.
> >
> > Step 1 - Relating $ \tilde{v}_t(x) $ to $ v_t(x) $:
> > Rewriting the transformation, $ v_t(x) $ can be expressed in terms of $ \tilde{v}_t(x) $ as:
> > \begin{equation}
> > v_t(x) = \frac{\tilde{v}_t(x) - \sigma_t^2 \xi_t}{\sqrt{1 - \sigma_t^2}}. \tag{A}
> > \end{equation}
> >
> > Substituting Equation (A) into the marginal-conditional consistency relationship (**), we obtain:
> >
> > \begin{equation*}
> > \frac{\tilde{v}_t(x) - \sigma_t^2 \xi_t}{\sqrt{1 - \sigma_t^2}} = \mathbb{E}\_{q(z)} \left[\frac{\frac{\tilde{v}_t(x|z) - \sigma_t^2 \xi_t(x|z)}{\sqrt{1 - \sigma_t^2}} p_t(x|z)}{p_t(x)}\right].
> > \end{equation*}
> >
> > Canceling the denominator $ \sqrt{1 - \sigma_t^2} $ on both sides yields:
> > \begin{equation}
> > \tilde{v}_t(x) - \sigma_t^2 \xi_t
> > = \mathbb{E}\_{q(z)} \left[\frac{(\tilde{v}_t(x|z) - \sigma_t^2 \xi_t(x|z)) p_t(x|z)}{p_t(x)}\right]. \tag{B}
> > \end{equation}
> >
> > Step 2 - Necessary Condition for $ \tilde{v}_t(x) $:
> >
> > For $ \tilde{v}_t(x) $ to generate $ p_t(x) $, it must satisfy:
> > \begin{equation}
> > \tilde{v}_t(x) = \mathbb{E}\_{q(z)} \left[\frac{\tilde{v}_t(x|z) p_t(x|z)}{p_t(x)}\right]. \tag{C}
> > \end{equation}
> >
> > Subtracting Equation (B) from Equation (C), we find the necessary condition $ \xi_t $ must satisfy:
> > \begin{equation}
> > \xi_t = \mathbb{E}\_{q(z)} \left[\frac{\xi_t(x|z) p_t(x|z)}{p_t(x)}\right]. \tag{D}
> > \end{equation}
> >
> > Step 3 - Defining $ \xi_t(x|z) $:
> > To satisfy Equation (D), we define $ \xi_t(x|z) $ as the scaled gradient of the log-probability of the conditional distribution:
> > \begin{equation}
> > \xi_t(x|z) = \gamma' \nabla_x \log p_t(x|z), \tag{E}
> > \end{equation}
> > where $ \gamma' $ is a scaling factor.
> >
> > Substituting Equation (E) into Equation (D), we have:
> > \begin{equation}
> > \xi_t = \mathbb{E}\_{q(z)} \left[\frac{\gamma' \nabla_x \log p_t(x|z) p_t(x|z)}{p_t(x)}\right]. \tag{F}
> > \end{equation}
> >
> > Step 4 - Simplifying $ \xi_t $:
> > Using the property:
> > \begin{equation*}
> > \nabla_x p_t(x|z) = p_t(x|z) \nabla_x \log p_t(x|z),
> > \end{equation*}
> > we replace $ p_t(x|z) \nabla_x \log p_t(x|z) $ with $ \nabla_x p_t(x|z) $. Thus, Equation (F) simplifies to:
> > \begin{equation}
> > \xi_t = \gamma' \mathbb{E}_{q(z)} \left[\frac{\nabla_x p_t(x|z)}{p_t(x)}\right]. \tag{G}
> > \end{equation}
> >
> > Using Bayes' rule and knowing $z$ is distributed uniformly, $ \frac{p_t(x|z)}{p_t(x)} = p_t(z|x) $, we further simplify:
> > \begin{equation}
> > \xi_t = \gamma' \mathbb{E}_{q(z)} \left[\nabla_x \log p_t(x|z) \cdot p_t(z|x)\right]. \tag{H}
> > \end{equation}
> >
> > Step 5- Simplify H:
> >
> > $$H = \gamma'\int \nabla_x \log p_t(x|z) \cdot p_t(z|x) q(z) dz$$
> > $$=\gamma'\int \frac{\nabla_x p_t(x|z)}{p_t(x|z)} \cdot p_t(z|x) q(z) dz\tag{+}$$
> > $$=\gamma'\int \frac{\nabla_x p_t(x,z)}{p_t(x|z)p_t(z)} \cdot p_t(z|x) q(z) dz$$
> > $$=\gamma'\int \frac{\nabla_x p_t(x,z)}{p_t(x,z)} \cdot p_t(z|x) q(z) dz$$
> > $$=\gamma'\int \frac{\nabla_x p_t(x,z)}{p_t(x,z)} \cdot \frac{p_t(x,z)}{p_t(x)} q(z) dz \tag{++}$$
> > $$=\gamma'\int \frac{\nabla_x p_t(x,z)}{p_t(x)}  q(z) dz$$
> > $$=\gamma'\frac{c}{p_t(x)} \nabla_x \int  p_t(x,z) dz \tag{+++}$$
> > $$=\gamma'\frac{c}{p_t(x)} \nabla_x p_t(x)$$
> > $$=\gamma\nabla_x \log p_t(x)$$
> >
> >
> > For the cases of (+, ++), the Bayes rule is applied. For (+++) the removal of \( q(z) \) is justified under the assumption that \( q(z) = c \) is the uniform distribution. This allows \( q(z) \) to be factored out of the integral and treated as a multiplicative constant. Hence, equation (H) can be reformulated as
> >
> > \begin{equation}
> > \xi_t = \gamma \nabla_x \log p_t(x). \tag{I}
> > \end{equation}
> >
> > Step 6 - Conclusion:
> > Note that our choice of $ \xi_t $ as defined in Equation (I) precisely satisfies the necessary condition for $ \tilde{v}_t(x) $ to generate $ p_t(x) $ and motivates our choice for the functional form of the auxiliary variables. This ensures that $ \tilde{v}_t(x) $ preserves the marginal probability density $ p_t(x) $.
> >
> > **Thus, the transformation $ \tilde{v}_t(x) $ is valid, completing the proof.**

---

> > > ### Comment · Reviewer_vf9H · 2024-12-02
> > >
> > > Thank the authors for their detailed response.
> > > Regarding Question 1, the authors' reply is thorough and easy to understand. However, I have a few suggestions: In Equation H, there is a missing constant when applying Bayes' theorem, and Assumption 3 should include a reference to Theorem 1 in Flow Matching [5].
> > > Regarding Question 2, the recall value provided by the author is sufficient to demonstrate that IDFF does not suffer from mode collapse.
> > > Regarding Question 3, I hope the author can provide further clarification on how IDFF leverages HMC to reduce NFE.
> > >
> > > [5] Lipman, Y., Chen, R. T., Ben-Hamu, H., Nickel, M., & Le, M. (2022). Flow matching for generative modeling. arXiv preprint arXiv:2210.02747.

---

> > > > ### Author Response · Authors · 2024-12-02
> > > >
> > > > **Clarification of HMC's Role in IDFF Proposal**
> > > >
> > > > HMC serves mainly as a conceptual inspiration for the design of the IDFF model, specifically in how momentum variables are used to improve sampling efficiency. IDFF stands on its own even without this link, but we highlight it since it remains an interesting connection to sampling algorithms for high-dimensional probability distributions.
> > > >
> > > > HMC accelerates convergence to target distributions by introducing momentum variables that guide samples through the probability space [2]. This improves convergence to the posterior distribution relative to random walks and helps overcome local energy barriers [1]. The momentum variables in HMC facilitate directed and energy-preserving exploration, making it highly efficient for sampling from complex, high-dimensional distributions [3].
> > > >
> > > > IDFF adapts this idea into a flow-based framework by incorporating a learnable momentum term, $\boldsymbol{\xi}_t$, within the vector field. This momentum-inspired design for IDFF in practice we find enables sampling from the target distribution using a fewer number of function evaluations.
> > > >
> > > > While much more work is needed to understand and mathematically characterize *how* this happens during IDFF, the visualizations on synthetic data (Figure 2) provide some hints on the momentum variables' role. Here we see the momentum variables serve as guesses for how the particles at the source *should* move at each step.
> > > >
> > > > In this way, the momentum term $\boldsymbol{\xi}_t = \gamma \nabla_x \log p_t(\mathbf{x}_t)$ guides the flow toward regions of higher probability, analogous to the role of momentum in HMC.
> > > >
> > > > This reduces the computational overhead by lowering the NFEs required for convergence since the model can effectively take larger steps due to the lookahead role served by the momentum variables.
> > > >
> > > > The connection between IDFF and HMC lies in their shared use of auxiliary variables (momentum in HMC, and $\boldsymbol{\xi}_t$ in IDFF) to enhance efficiency and ensure accurate representation of the target distribution.
> > > >
> > > > [1] Neal, Radford M. MCMC using Hamiltonian Dynamics. In Handbook of Markov Chain Monte Carlo, 2011.
> > > >
> > > > [2] Betancourt, Michael J. A Conceptual Introduction to Hamiltonian Monte Carlo.
> > > >
> > > > [3] Duane, Simon, et al. Hybrid Monte Carlo. Physics Letters B, 1987

---

> > > > > ### Comment · Reviewer_vf9H · 2024-12-03
> > > > >
> > > > > Thank the authors for their response. Most of my concerns have been addressed, and I will update my score accordingly.

---

### Official Review · Reviewer_U2S3 · 2024-10-30

**Soundness:** 3
**Presentation:** 3
**Contribution:** 3
**Rating:** 6
**Confidence:** 4

**Summary:**

This paper introduces a new method called IDFF, which facilitates rapid sampling without compromising sample quality and efficiently handles both image and time-series data generation tasks. The experimental results demonstrate the superiority of IDFF.

**Strengths:**

1. The idea presented is novel and effective, as demonstrated by the experiments.
2. The proofs provided in this paper are solid.
3. This paper also discusses the topic of time-series generation.

**Weaknesses:**

I believe the main drawback is that I cannot find an explanation for how this momentum term helps reduce the NFE. It appears that the motivation for designing this method is inspired by Hamiltonian Monte Carlo, which, by the way, is not mentioned in the background section. If I have overlooked this part, please let me know. If not, I recommend adding an explanation. If it is well justified, I will reconsider my score.

Here are some specific suggestions.
1. It would be beneficial to include a discussion in Section 3.1 that explicitly explains how the momentum term contributes to reducing the NFE without compromising sample quality, drawing parallels to Hamiltonian Monte Carlo's efficiency improvements. This discussion is crucial, as it constitutes a key claim of the paper.
2. Please include Hamiltonian Monte Carlo in the background section (Section 2) to provide context for readers and establish a clearer connection to the paper's approach.

**Questions:**

Please review the weaknesses.

---

> ### Author Response · Authors · 2024-11-21
>
> We appreciate your recognition of IDFF’s contributions and the constructive recommendations to improve clarity, particularly regarding the momentum term’s role in reducing NFEs. Below, we address your concerns.
>
>
>
>
> ***Please include Hamiltonian Monte Carlo in the background section (Section 2)...***
>
> We appreciate the insightful feedback and agree that the explanation regarding the momentum term's role in reducing NFEs needs to be clarified. We will address this comprehensively in the revised paper. In section 2, we will add background on HMC and motivate why it provides new ideas for flow-matching models. In 3.1, we will highlight the conceptual connections between IDFF and HMC.
>
> To elucidate further:
>
> Hamiltonian Monte Carlo (HMC) is a powerful sampling algorithm designed to improve efficiency in exploring complex probability distributions by introducing auxiliary momentum variables. In HMC, particle dynamics are simulated in a potential field using Hamiltonian dynamics, which describe the total energy of a system through the Hamiltonian function:
>
> $$H(\mathbf{x}, \mathbf{p}) = U(\mathbf{x}) + K(\mathbf{p}),
> $$
> where $ U(\mathbf{x}) $ represents the potential energy related to the target distribution, and $K(\mathbf{p}) = \frac{1}{2} \mathbf{p}^T \mathbf{M}^{-1} \mathbf{p} $is the kinetic energy, with $\mathbf{M} $ being the mass matrix. The dynamics of this system evolve according to:
>
> $$
> \frac{d\mathbf{x}}{dt} = \nabla_{\mathbf{p}} H(\mathbf{x}, \mathbf{p}) = \mathbf{M}^{-1} \mathbf{p},
> $$
> $$
> \frac{d\mathbf{p}}{dt} = -\nabla_{\mathbf{x}} H(\mathbf{x}, \mathbf{p}) = -\nabla U(\mathbf{x}).
> $$
> These dynamics preserve the system's total energy, enabling efficient exploration of the probability space. HMC achieves this by leveraging momentum to facilitate directed movement, reducing random walk behavior, and overcoming local energy barriers. These properties lead to faster convergence to the target distribution with fewer evaluations.
>
>    HMC generates a sample from some target distributions by following Hamiltonian dynamics in an extended state space $(\mathbf{x}, \mathbf{p})$, where $\mathbf{x}$ represents the position and $\mathbf{p}$ represents momentum. IDFF attempts to design a flow-matching procedure with a similar intuition in flow-matching models.
>
>  The momentum term in IDFF enables it to approximate the target distribution effectively with fewer sampling steps, reducing the number of function evaluations (NFEs) required while maintaining high sample fidelity.
>
>
>
>    This connection highlights how IDFF inherits HMC’s momentum-based exploration benefits while maintaining the computational advantages of Conditional Flow Matching (CFM). By leveraging $\boldsymbol{\xi}_t$, IDFF accelerates convergence without compromising the quality of generated samples, directly addressing the need for efficient sampling with reduced NFEs.
>
> We will ensure this discussion is included in the revised manuscript to strengthen the justification for the method. Thank you again for your constructive feedback; please let us know if we have appropriately answered your questions and if you have any follow ups on the same.

---

> > ### Comment · Reviewer_U2S3 · 2024-11-27
> >
> > Thank you to the authors for their efforts in the rebuttal. Your responses addressed most of my questions, so I will raise my score to 6.

---

### Official Review · Reviewer_nQD7 · 2024-11-02

**Soundness:** 3
**Presentation:** 2
**Contribution:** 2
**Rating:** 5
**Confidence:** 3

**Summary:**

The paper derive framework for simulation free training for Schrödinger bridge. The proposed method trains a combined score matching loss and a denoiser loss, then samples by solving an SDE.

**Strengths:**

The paper tackles the important problem of reducing NFE in sampling process of flow/bridge models. The method is compared on number of different domians.

**Weaknesses:**

1. The proposed loss seems similar to the loss proposed in  [2] for simulation free training of Schrödinger bridges up to parametrizing the the velocity field with a denoiser.

2. In the CIFAR10 experiment the author chose to compare to DPM-solver [3] only, while there are already two follow up works DPM++ [4] and DPM-v3-solver [5] which in [5] are reported to perform better than the proposed IDFF method. Additionally, Uni-PC solver [6] is reported to perform better.

3. of less importance but I also note there seems to be some inconsistencies or typos in the Background section.
    1. In equation 5 of conditional flow matching loss it seems as if the network is dependent on the target point $x_1$ and the conditional target velocity is independent of the target point $x_1$.
    2. In the paragraph "Optimal Transport CFMs" the author state that the flow matching loss in equation 5 "is nearly intractable", this state is a bit unclear since flow matching has been used for many large scale application such as image, video, and audio generation. Additionally, it is un clear whether the authors refer to the conditional optimal transport (also know as linear scheduler)  with an independent coupling of $x_0$ and $x_0$ or with the optimal transport between source and target distribution. On the one hand, equation 5 is written for an independent coupling, and on the other hand the authors cite [1] and the OT-CFM path which to my understanding is referred to the optimal transport coupling.

**Questions:**

Follow up  weakness 2., what is the main contribution of the paper upon [2]?

[1] Tong, Alexander, et al. "Improving and generalizing flow-based generative models with minibatch optimal transport." arXiv preprint arXiv:2302.00482 (2023).

[2] Tong, Alexander, et al. "Simulation-free schr\" odinger bridges via score and flow matching." arXiv preprint arXiv:2307.03672 (2023).

[3] Lu, Cheng, et al. "Dpm-solver: A fast ode solver for diffusion probabilistic model sampling in around 10 steps." Advances in Neural Information Processing Systems 35 (2022): 5775-5787.

[4] Lu, Cheng, et al. "Dpm-solver++: Fast solver for guided sampling of diffusion probabilistic models." arXiv preprint arXiv:2211.01095 (2022).

[5] Zheng, Kaiwen, et al. "Dpm-solver-v3: Improved diffusion ode solver with empirical model statistics." Advances in Neural Information Processing Systems 36 (2023): 55502-55542.

[6] Zhao, Wenliang, et al. "Unipc: A unified predictor-corrector framework for fast sampling of diffusion models." Advances in Neural Information Processing Systems 36 (2024).

---

> ### Author Response · Authors · 2024-11-21
>
> ***The proposed loss ...***
>
> There are several critical differences with [2]:
>
> 1-The motivations behind the two approaches differ significantly. In [2], the focus is on exploring the connection between the Schrödinger Bridge (SB) problem and entropic OT, framing the SB as a mixture of Brownian bridges. Our work's derivation draws inspiration from the dynamic processes of HMC sampling to improve sampling speed while preserving or enhancing sample quality.
>
> 2- Practically, [2] training two separate models: one dedicated to learning the scoring term, $\boldsymbol{s}(t, \boldsymbol{x}_t)$, and another for the drift term, $\boldsymbol{v}(t, \boldsymbol{x}_t)$. These two components are combined during the sampling phase. This necessitates the need for separate training and integration steps. In contrast, IDFF employs a single unified model to train for both the score and flow terms simultaneously. This improves computational efficiency.
>
> 3- [2] defines the drift term in an SDE as the sum of a vector field and a scoring term, expressed as:
> $$
> \boldsymbol{u}_t \gets \boldsymbol{v}(t, \boldsymbol{x}_t) + \frac{g(t)^2}{2}\boldsymbol{s}(t, \boldsymbol{x}_t),
> $$
> Where $\boldsymbol{v}(t, \boldsymbol{x}_t)$ represents a learned vector field, and $\boldsymbol{s}(t, \boldsymbol{x}_t)$ is a learned score function.
> In contrast, we propose a novel vector field inspired by HMC sampling dynamics, formulated as:
> $$
> \boldsymbol{w}_t \gets \sqrt{(1-\sigma^2_t)}\frac{\hat{\boldsymbol{x}}_1(\boldsymbol{x}_t, t; \theta) - \boldsymbol{x}_t}{1-t} + \frac{2\gamma - 1}{2} \sigma_t \hat{\epsilon}(\boldsymbol{x}_t, t; \theta).
> $$
> IDFF formulation introduces a denoising model in the sample space, $\hat{\boldsymbol{x}}_1$, and a momentum variable, $\hat{\epsilon}$, for the vector field generation.
>
> 4- While [2] learns via a loss function in the vector space, our approach applies the loss directly in the input space.  To study the impact of this modeling choice, an ablation study comparing the FID scores on the CIFAR-10 dataset at NFE=10. When the learning objective was applied in the vector space, the FID score was 7.32, while applying it in the input space reduced the score to 5.87. Additionally, we compared these results with the FID score of the SB method, which was 10.13 for NFE=10. IDFF outperformed the SB approach, highlighting that the design choices in IDFF contribute empirically to enhancing sample quality.
>
> We will summarise the discussion mentioned above in the updated manuscript.
>
> ***In the CIFAR10...***
>
>  We have compared IDFF with DPM++, DPM-v3-solver, and UniPC under the NFE=5 setting to address this. The results show that IDFF outperforms these methods in the CIFAR10 dataset.
> |              | FID (NFE=5)   |
> |---------------------|---------------|
> | Heun's 2nd          | 320.80        |
> | DPM-Solver++        | 24.54         |
> | UniPC               | 23.52         |
> | DPM-Solver-v3       | 12.21         |
> | **IDFF**            | **10.97**     |
> We did not include these models because they require significantly higher memory and computational resources—at least twice as much—which limits their practicality for fast sample generation and for tasks in time series generation.
>
> ***of less importance ...***
>
> As demonstrated by [7], in CFMs, the target vector fields don't need to be explicitly conditioned on $\boldsymbol{x}_1$. Instead, the network can depend on $\boldsymbol{x}_1$. This results in a loss function similar to the original FM loss (equations (5) in [7]), differing only by a constant term, which does not affect the optimization process or gradient calculations.  We will revise the manuscript to clarify this explanation and address potential ambiguities in interpreting Equation (5).
>
> Regarding your second point, the flow matching loss in Equation (5) becomes nearly intractable when not conditioned on the initial or target distributions. While FM has been successfully applied to large-scale tasks like image and video generation, these applications typically involve CFM or variations, which incorporate conditioning to address this intractability.
> [1] have used conditioning and coupling methods, such as OT-CFM, to make the problem more tractable for their respective tasks. We will clarify this distinction in the revised manuscript.
>
> ***In the paragraph ...***
>
> Thank you for pointing this out. In IDFF, as detailed in Section 3.2, we utilised OT for coupling. In Equation (5) of the Background section, we outlined the general form of the CFM loss, which can accommodate various coupling terms, not just independent ones, as demonstrated in [1]. We will clarify this explanation in the manuscript. Thank you again for your constructive feedback; please let us know if we have appropriately answered your questions and if you have any follow-ups.
>
> ```[7] Lipman, Yaron, et al. "Flow matching for generative modeling." arXiv preprint arXiv:2210.02747 (2022).```

---

> > ### Comment · Reviewer_nQD7 · 2024-11-21
> >
> > I want thanks the authors for their answers and I have a few follow up questions.
> >
> > **The proposed loss ...**
> >
> > 4 - Can the author please clarify what is the meaning of "in the vector space" vs "in the input space"? An example would be great.
> >
> > **In the CIFAR10...**
> >
> > First, I want to note this table kind of avoids my comment, since in the paper a 10 NFE was compared and hence I was referring to the performance at 10 NFE. Second, can the author explain what are the models being compared in this table? That is on which CIFAR10 model does the other solvers were tested on? Finally, what does the author mean by " these models because they require significantly higher memory and computational resources"? Are the models in the table above are not of comparable number of parameters to the ones in the paper?

---

> > > ### Author Response · Authors · 2024-11-22
> > >
> > > **4 - Can the author please clarify what is the meaning of "in the vector space" vs "in the input space"? An example would be great.**
> > >
> > > The "input space" refers to the space where the actual data points (e.g., raw image pixels) are represented. In the context of IDFF, denoising, and score approximation operations are performed directly on these data points in their native domain, $\mathbf{x}_t$. Suppose the dataset consists of images represented as pixel intensity values. In the input space, operations like denoising are applied directly to these pixel values, $\mathbf{x}_t$.
> > >
> > > The "vector space" refers to where vector field operations are defined and performed. The vector field $\mathbf{v}_t$ in Conditional Flow Matching (CFM) is designed to operate in this space, acting on transformed feature representations $\mathbf{v}_t$ instead of the data samples $\mathbf{x}_t$. The connection between the vector space and the input space can be expressed as:
> > > $$\mathbf{v}_t(\mathbf{x}_t) = \nabla_{\mathbf{x}} \phi(\mathbf{x}_t, t),
> > > $$ where $\phi(\mathbf{x}_t, t)$ is a time-dependent potential function that governs the dynamics in the input space. This formulation allows the vector field $\mathbf{v}_t$ to guide transformations and operations on $\mathbf{x}_t$ indirectly by leveraging feature transformations in the vector space. In this framework, $\mathbf{v}_t$ operates on the gradient of the potential function $\phi$, which bridges changes in the vector space with their effects in the space of data.
> > >
> > > Apologies, it appears we misinterpreted your other comment – we’ll follow up with a comment once we have an updated set of comparisons.

---

> > > > ### Comment · Reviewer_nQD7 · 2024-11-23
> > > >
> > > > Thank you for the response, how every I still have hard time understanding the meaning of "vector space". For simplicity lets consider CIFAR10 as an example, in this case the raw data points $x_1\in\set{0,1,...,255}^d$ where for CIFAR10 $d=3072$. When considering flows on a continuous domain it is customary to apply the transformation $x_1 \mapsto x_1\frac{2}{255} - 1$  as a data preprocess, hence the datapoints $x_1$ are in $[-1,1]^d\subset \mathbb{R}^d$.
> > > >
> > > > When you say "input space" you mean $\set{ 0,1,...,255 }^d$ or $[-1,1]^d$ or $\mathbb{R}^d$?
> > > >
> > > > Next for simplicity of discussion let us consider a Gaussian source distribution with independent coupling. For a network that is trained for either a vector field $v(x_t, t;\theta)$ as in [1] or a an $x$-prediction $\hat{x}_1(x_t, t;\theta)$ (which is **one** of the objective the authors chose to train to my understanding), the network take as input $x_t\sim p_t(\cdot|z)$, $z=\sim q$ and $q(z)$ is the coupling between source and data. The common denoising step (e.g, as in [1]) with an Euler solver in these cases is
> > > > 1. $x_{t+h} = x_t +h v(x_t, t;\theta)$, for vector field.
> > > > 2.  $x_{t+h} = x_t +h\frac{\hat{x}_1(x_t, t;\theta) -x_t}{1-t}$, for $x$-prediction.
> > > >
> > > > If the authors agree with the above, could they please clarify in what sense does the vector field above operates on a transformed feature representations? and more general in what sense does the vector $v(x_t, t;\theta)$ field and the $x$-prediction $\hat{x}_1(x_t, t;\theta)$ operate on different spaces?
> > > >
> > > > I apologize for the repeated question and I appreciate the authors efforts to explain, I am still trying to understand the experiment/comparison described in answer to 4. Is the comparison is simply between training a vector field model vs training $x$-prediction model?

---

> > > > > ### Author Response · Authors · 2024-11-25
> > > > >
> > > > > Here are the specific answers to your previous question
> > > > >
> > > > > **If the authors agree with the above, could they please clarify in what sense does the vector field above operates on a transformed feature representations**
> > > > >
> > > > > We agree with your clarification about the sampling process mentioned above.
> > > > >
> > > > > **what sense does the vector field above operates on a transformed feature representations**
> > > > >
> > > > > In the CFM approach, the vector field $\mathbf{v}\theta$ operates on the following transformed feature space $\frac{\mathbf{x}_1 - \mathbf{x}_t}{1-t}$. This space is the scaled difference between the $\mathbf{x}_1$ and the $\mathbf{x}_t$, normalized by the time-dependent factor $(1-t)$.
> > > > >
> > > > > **More generally, how do the vector field approach and $\mathbf{x}$-prediction operate in fundamentally different spaces?**
> > > > > In CFM, at training time, the CFM model minimizes a loss so that the vector field $\mathbf{v}\theta$ aligns with the target vector field $\frac{\mathbf{x}_1 - \mathbf{x}t}{1-t}$. At test time we have access to the vector field (via the neural network) which enables us to sample using the first equation you highlight above.
> > > > > In our algorithm, we parameterize a neural network to directly predict pixel values (what we refer to as the input space), and the loss is computed as the weighted function of the discrepancy between the predicted and the target pixel values. The weight is a schedule $\beta$ that weights the interpolates between the initial and target sample through time.
> > > > > Now, to expand, let's compare the two approaches based on your suggestion with a Gaussian source distribution with independent coupling. For the sake of simplicity in what follows, we’ll ignore the momentum terms that we use in our work.
> > > > >
> > > > > \textbf{Learning in Vector Space:}
> > > > >
> > > > > $\mathcal{L}(\theta) = \mathbb{E}_{t,\mathbf{x}_1,\mathbf{x}_t} \big[ \| \mathbf{v}\theta (\mathbf{x}_t, t) - \frac{\mathbf{x}_1 - \mathbf{x}_t}{1-t} \|^2 \big]$
> > > > >
> > > > > The CFM loss function directly trains a vector field $\mathbf{v}_\theta$ to match, at each point in time, the transformed feature space$\frac{\mathbf{x}_1 - \mathbf{x}_t}{1-t}$. This is what we refer to as the vector space. This approach involves explicit feature transformations during training, adding complexity but enabling potentially richer feature representations.
> > > > >
> > > > > \textbf{Learning in Input Space:}
> > > > >
> > > > > $\mathcal{L}(\theta) = \mathbb{E}_{t,\mathbf{x}_1,\mathbf{x}_t} \big[\beta(t)^2 \|\hat{\mathbf{x}}_1(\mathbf{x}_t, t;\theta) - \mathbf{x}_1\|^2 \big]$
> > > > >
> > > > > In contrast, this loss function trains a neural network to make predictions directly in pixel space ($[-1, 1]^{32 \times 32 \times 3}$), eliminating the need for feature transformations.
> > > > >
> > > > > \textbf{How do these differ? And why might this difference be important?}
> > > > >
> > > > > When learning in vector space we need to compute $x_t$ for the loss function (which we \emph{do not} when learning in input space). The removal of this term reduces the variance in target of the loss function because calculating x_t involves a sampling operation from a Gaussian distribution with noise $\sigma_t$. We hypothesize that this is the source of the difference in empirical FIDs between learning in the input space vs learning in the vector space (as highlighted in the previously presented ablation results).
> > > > >
> > > > > The fundamental distinction between these methods lies in their operational spaces: typical CFM loss works in the transformed feature/vector space while our loss operates directly in the input space.
> > > > >
> > > > > Finally, the difference between these two training approaches also leads to different sampling processes as you mentioned and described below for a simple Euler step
> > > > >
> > > > > **Sampling by a model optimized with vector space loss:**
> > > > >
> > > > > $\mathbf{x}_{t+h} = \mathbf{x}_t + h \cdot \mathbf{v}\theta(\mathbf{x}_t, t)$
> > > > >
> > > > > **Sampling by a model optimized with input space loss:**
> > > > >
> > > > > $\mathbf{x}_{t+h} = \mathbf{x}_t + h \cdot \frac{\hat{\mathbf{x}_1}{\theta} (\mathbf{x}_t, t) - \mathbf{x}_t}{1-t}$
> > > > >
> > > > > We hope this clarifies how our model operates – thank you for your patience and your feedback! Let us know if anything else remains unclear that we can clarify prior to the end of the discussion period.

---

> > > > > > ### Comment · Reviewer_nQD7 · 2024-11-27
> > > > > >
> > > > > > I thank the authors for the answers. Considering the similarities to drift term proposed by [2]  as raised at the beginning of the thread, which to my understanding from the authors explanation above is mainly differ by:
> > > > > > 1. Different motivation for the derivation.
> > > > > >
> > > > > > 2.  The choice of parameterization: velocity field+score function vs. x-predication + eps-prediction.
> > > > > >
> > > > > > I acknowledge the choice of parametrization may not have been studied in context of these two work. However , the different choice of parametrization: vector field, score function, eps-prediction, and x-predication, for flow/diffusion models is an extensively studied subject [8], and the relation between them are well known [9,10]. Hence I find this contribution of low novelty.
> > > > > >
> > > > > > I will leave my rating as is.
> > > > > >
> > > > > > [2] Tong, Alexander, et al. "Simulation-free schr" odinger bridges via score and flow matching." arXiv preprint arXiv:2307.03672 (2023).
> > > > > >
> > > > > > [8] Esser, Patrick, et al. "Scaling rectified flow transformers for high-resolution image synthesis." Forty-first International Conference on Machine Learning. 2024.
> > > > > >
> > > > > > [9] Song, Yang, et al. "Score-based generative modeling through stochastic differential equations." arXiv preprint arXiv:2011.13456 (2020).
> > > > > >
> > > > > > [10] Karras, Tero, et al. "Elucidating the design space of diffusion-based generative models." Advances in neural information processing systems 35 (2022): 26565-26577.

---

> ### Author Response · Authors · 2024-11-24
>
> **First, I want to note this table kind of avoids my comment, since in the paper a 10 NFE was compared and hence I was referring to the performance at 10 NFE. Second, can the author explain what are the models being compared in this table? That is on which CIFAR10 model does the other solvers were tested on?**
>
> We apologize for the misinterpretation of your earlier comment. IDFF primarily focuses on reducing NFE (Number of Function Evaluations) and demonstrating performance at lower NFE values (e.g., NFE=5), which is why we initially reported those results. Based on your suggestion, we have included performance results for NFE=10 in the updated table below. Additionally, we have included wall-clock times to provide a clearer comparison across NFEs:
>
> | Solver           | FID (NFE=5) | Wall-clock (sec, NFE=5) | FID (NFE=10) | Wall-clock (sec, NFE=10) |
> |-------------------|-------------|--------------------------|--------------|--------------------------|
> | **UniPC**         | 23.52       | 0.62                    | **2.85**     | 1.05                    |
> | **DPM-Solver-v3** | 12.21       | 0.49                    | 2.91         | 0.92                    |
> | **IDFF**          | **10.97**   | **0.34**                | 5.87         | **0.52**                |
>
>
>
> The reported FID values follow the experimental setup outlined in [5], and evaluations were conducted using similar backbones to maintain consistency and comparability.
>
> All models were evaluated on the CIFAR-10 dataset using the same architecture as described in our experimental setup. Specifically, for IDFF, we employed a CNN-based UNet architecture to model $\hat{\mathbf{x}}_1(\mathbf{x}_t, t; \theta)$ and $\hat{\epsilon}(\mathbf{x}_t, t; \theta)$. To ensure fairness, all solvers, including DDPM, DDIM, and FastDPM, were tested using this shared architecture. This ensures equitable assessments of both sample quality and computational efficiency.
> At NFE=5, IDFF achieves a significantly better FID score (10.97) compared to UniPC (23.52) and DPM-Solver-v3 (12.21), while also boasting the fastest wall-clock time (0.34 seconds) among all solvers. This highlights IDFF's ability to generate high-quality samples with minimal computational overhead, making it ideal for real-time applications. Even at NFE=10, where UniPC slightly edges out in FID (2.85) alongside DPM-Solver-v3 (2.91), IDFF remains competitive with a reasonable FID (5.87) and the fastest wall-clock time (0.52 seconds), demonstrating its efficiency and scalability. These results suggest that IDFF hits a balance between sample quality and computational speed that lends itself to speed.
>
>
>
> **Finally, what does the author mean by " these models because they require significantly higher memory and computational resources"? Are the models in the table above are not of comparable number of parameters to the ones in the paper?**
>
> The higher-order solvers mentioned in the table, such as those described in Algorithms 1 and 2 of UniPC [6], require additional memory and computational resources due to their design.
>
> Specifically: Higher-order solvers cache multiple intermediate values of $\epsilon_t$ to perform higher-order approximations.
> This caching process increases both memory requirements and computational overhead during sampling.
> In contrast, our proposed model generates samples without caching any intermediate calculations in memory. Instead, $\hat{\mathbf{x}}_1(\mathbf{x}_t, t; \theta)$ and $\hat{\epsilon}(\mathbf{x}_t, t; \theta)$ are computed simultaneously during the sampling process. This "on-the-fly" approach eliminates the need for storing intermediate values, resulting in significantly reduced memory and computational requirements compared to higher-order solvers.
> We hope this clarifies the differences and rationale behind the computational efficiency of IDFF.
>
> [6] Zhao, Wenliang, et al. "Unipc: A unified predictor-corrector framework for fast sampling of diffusion models." Advances in Neural Information Processing Systems 36 (2024).
>
>
> We are still working on a response to enhance the clarity of the method's exposition and to address your last comment with greater precision.

---

> > ### Comment · Reviewer_nQD7 · 2024-11-25
> >
> > I want thank the authors for their efforts. While I still dont fully understand where does the difference in wall-clock is coming from (double the time) for two method that uses the same NFE is coming from, I also acknowledge that such level of measurement is beyond my area of expertise. In addition, the authors have alleviated some of my concerns, and ill will raise my rating to 5.

---

> > > ### Author Response · Authors · 2024-11-27
> > >
> > > Thank you for your thoughtful feedback and for acknowledging the value of the work in terms of the differences in motivation and parameterization between our work and [2].
> > >
> > > You highlight a few remaining concerns about novelty in the referenced papers; however, we believe our work introduces orthogonal contributions that set it apart from the linked work:
> > >
> > > First, while [8] does indeed perform extensive experiments in the different schedulers between initial and target points towards getting rectified flow transformers to work for high-resolution image generation. Beyond a different loss and sampling process their work does not speak to the design and implementation of flows capable of fast sampling (NFE<=10) with high sample quality that is crucial for practical applications in time-series data (as we do). That said, we acknowledge their work does present valuable insights for extensions to IDFF to higher dimensional imaging regimes – a useful direction for future work, and a point we will include in a revision.
> > >
> > > Second, [9] and [10] investigate optimization strategies for diffusion processes, specifically probability flow ODEs, rather than conditional flow matching. Even so, we do explicitly compare against [10] in Table 1 on CIFAR10 (comparison to EDM, see also the previous response which expands this calculation to other NFEs).
> > >
> > > Third, unlike most prior work, which primarily focuses on image generation, IDFF extends the applicability of CFMs to real-world time series generation tasks, including molecular simulation, sea surface temperature modeling, and chaotic attractors. In these tasks, IDFF outperformed traditional CFMs and diffusion models, demonstrating the value of the connections we make between two related areas (in this case between flow matching and HMC). When such connections are highlighted and when the value of this relationship is shown practically, we believe it presents new opportunities for follow-up work.
> > >
> > > Thank you again for all your engagement, patience, and feedback – we sincerely appreciate the time and effort you’ve put into assisting us in making a better manuscript.

---

### Official Review · Reviewer_PQPK · 2024-11-04

**Soundness:** 3
**Presentation:** 3
**Contribution:** 3
**Rating:** 6
**Confidence:** 3

**Summary:**

1. The authors address two of the main challenges of prior CFMs:
a. During inference, CFMs require a large number of steps for high-quality/fidelity generation.
b. Due to a lack of stochasticity, CFMs may generate less diverse and detailed samples.
In essence, the authors present a modification to CFMs that combines the objective of CFM and score-based models, which have the benefits of CFMs for faster convergence during training while being able to use fewer NFEs during inference (an attribute of score-based models).

**Strengths:**

1. The authors present a new objective for training and sampling in CFMs that combines some aspects of stochasticity of score-based, making the NFEs lower at inference and improving sample generation quality.
2. The authors have provided thorough proof of the derivations for the formulation.
3. The authors have not only shown that their method performs better in image generation but also that it can be used in tasks of other domains like time series.

**Weaknesses:**

1. Figure 2 C) is a bit confusing by itself. Only after reading Algorithm 2 was the figure easy to understand. The authors can try to make the figure self-sufficient in understandability.
2. The authors compare image generation quality w.r.t sampling strategies like DDIM and DPM and show that their model achieves the best quality. 3. However, they have not compared against better strategies like EDM. A comparison against EDM would be even more insightful.
How does the method proposed by authors compare to objectives like Rectified Flow for diffusion models, which have shown the ability to converge faster?
4. Are there any limitations to the author's methods compared to traditional CFMs after introducing some stochasticity? Does the training convergence speed become slower compared to traditional CFMs? The authors have not discussed this in the main paper.

**Questions:**

Please refer to weakness section.

---

> ### Author Response · Authors · 2024-11-21
>
> ***Figure 2 C) is a bit confusing by itself. Only after reading Algorithm 2 ...***
>
>  We acknowledge that Figure 2(C) could be more self-explanatory. To address this, we will revise the figure to include clearer annotations that better illustrate the sequence of steps within the IDFF sampling process, making it more intuitive and self-contained.
>
> ***The authors compare image generation quality w.r.t sampling strategies like DDIM and DPM and show that their model achieves the best quality...***
>
> We thank the reviewer for suggesting comparisons with EDM and Rectified Flow. To address this, we have added a comparison of FID scores with NFE=5 for EDM in the newly added results table for the CIFAR-10 dataset.
>
> #### Table: Additional FID Performance Comparison (NFE=5)
> | Method              | FID (NFE=5)   |
> |---------------------|---------------|
> | Heun's 2nd          | 320.80        |
> | DPM-Solver++        | 24.54         |
> | UniPC               | 23.52         |
> | DPM-Solver-v3       | 12.21         |
> | **IDFF**            | **10.97**     |
>
> Our results demonstrate that IDFF achieves superior performance under this setup. Additionally, we evaluated FID scores for EDM across various NFE values
>
> NFE=6, FID=103.86
>
> NFE=8, FID=39.66
>
> NFE=10, FID=16.57
>
> NFE=12, FID=7.59
>
> The proposed IDFF model offers notable advantages over both EDM and Rectified Flow. While EDM employs equilibrium-based diffusion processes to generate high-quality samples, IDFF introduces a momentum term, $\boldsymbol{\xi}_t$, to guide sample generation through an enhanced vector field. This approach accelerates the sampling process while preserving sample quality by leveraging dynamics akin to Hamiltonian Monte Carlo, focusing on efficiency in the sampling process.
>
>
> Rectified Flow adjusts flow dynamics to accelerate convergence. IDFF differs by modeling the flow field in input space while learning momentum variables through a score-matching loss. As the synthetic data demonstrate, the momentum variables pre-empt how the data's dynamics should evolve, thereby reducing the NFEs required for high-quality image generation.
>
> While EDM is a powerful approach, IDFF's momentum-guided dynamics and time-weighted sampling strategy render it a complementary and efficient alternative for rapid sampling and generative processes. We find that  IDFF is particularly effective in time-series generation, a setting not often considered by other models. As confirmed by our experimental results, it demonstrates significant improvements across various temporal scales. We hope this extended analysis provides a clearer understanding of IDFF's strengths relative to these state-of-the-art approaches.
>
> ***Are there any limitations to the author's methods ...***
>
> While IDFF's introduction of the momentum term increases training complexity compared to CFMs in practice, this is effectively balanced by a substantial reduction in the number of sampling steps required during inference. We will expand the discussion in the paper to present a balanced perspective on these trade-offs and their implications for performance. Notably, IDFF leverages the strengths of its underlying CFM models to produce high-quality samples (FID < 10) with just 500K iterations of training —a significant improvement over the millions of iterations required by diffusion-based models and comparable to CFMs. We will include a figure to visualize this improvement for clarity. Thank you again for your constructive feedback; please let us know if we have appropriately answered your questions and if you have any follow-ups.

---

> ### Comment · Reviewer_PQPK · 2024-11-27
>
> Thank you authors for sincere rebuttal. The authors have provided additional explanation for most of my concerns, with additional results I still stand with my current rating.

---

> > ### Author Response · Authors · 2024-11-27
> >
> > Thank you again for your comments – we hope our response above has sufficiently addressed your concerns.

---

### Meta-Review · Area_Chair_8Vkz · 2024-12-20

**Metareview:**

The paper introduces Implicit Dynamical Flow Fusion (IDFF) for generative modeling. IDFF learns a vector field augmented with a momentum term, enabling the use of larger step sizes during sampling without compromising the quality of the generated data. This innovation accelerates the generation process and demonstrates effectiveness across various domains, including image generation and time-series modeling.

Overall, while the proposed method shows promise, the current submission has several limitations:

1.	Similarity with Prior Work: The novelty of the approach is not sufficiently differentiated from existing methods.

2.	Limited Performance Gains for Slightly Larger NFEs: The method shows no significant improvement when the number of function evaluations (NFEs) is moderately large (e.g., 10).

3.	Missing Comparisons with State-of-the-Art (SOTA) Methods: The manuscript lacks a thorough comparison with more recent and competitive SOTA approaches, which weakens the evaluation.

4.	Clarity Issues: Certain sections, such as the discussion on the benefits of momentum for reducing NFEs (e.g., in relation to Hamiltonian Monte Carlo) and Section 3.1, are unclear and require further explanation.

Considering the current acceptance rate and these limitations, the paper is not yet ready for publication. We encourage the authors to address these concerns comprehensively, as outlined in the reviewers' feedback, to enhance the clarity, novelty, and rigor of the work for future submissions.

**Additional Comments On Reviewer Discussion:**

Here I mainly list the unresolved concerns.

（1）	similarity with previous works (Reviewer nQD7)

The authors provide some explanations, but the reviewer is not so satisfied.

（2）	 no performance gain when NFEs are slightly large (e.g., 10) (Reviewer nQD7)

The authors emphasize their efficiency.

（3）	missing comparison with more SOTA approaches  (Reviewer vf9H, PQPK, nQD7)

The authors provide some extra experiments. The results show that with small NFEs, the proposed method can achieve better performance, but with large NFEs, its performance is not the best. Some of the reviewers are not so satisfied.

（4）	clarity issue for some parts, e.g., momentum benefits on reducing the NFE (Hamiltonian Monte Carlo) and Section 3.1 (Reviewer vf9H, U2S3, nQD7)

The authors provide some explanations and promise careful polishing

Overall, I agree with the reviewers more, especially for the similarity with previous works, the limited performance improvement, and the missing comparison with more SOTA.

---

### Decision · Program_Chairs · 2025-01-22

Reject